# EZH2 depletion potentiates MYC degradation inhibiting neuroblastoma and small cell carcinoma tumor formation

Liyuan Wang[1,2], Chan Chen[2], Zemin Song[3], Honghong Wang[3], Minghui Ye[2], Donghai Wang[2], Wenqian Kang[2], Hudan Liu 🄳 [2] & Guoliang Qing 🄳 [1,2 ✉]

Efforts to therapeutically target EZH2 have generally focused on inhibition of its methyltransferase activity, although it remains less clear whether this is the central mechanism whereby EZH2 promotes cancer. In the current study, we show that EZH2 directly interacts with both MYC family oncoproteins, MYC and MYCN, and promotes their stabilization in a methyltransferase-independent manner. By competing against the SCF^FBW7 ubiquitin ligase to bind MYC and MYCN, EZH2 counteracts FBW7-mediated MYC(N) polyubiquitination and proteasomal degradation. Depletion, but not enzymatic inhibition, of EZH2 induces robust MYC(N) degradation and inhibits tumor cell growth in MYC(N) driven neuroblastoma and small cell lung carcinoma. Here, we demonstrate the MYC family proteins as global EZH2 oncogenic effectors and EZH2 pharmacologic degraders as potential MYC(N) targeted cancer therapeutics, pointing out that MYC(N) driven cancers may develop inherent resistance to the canonical EZH2 enzymatic inhibitors currently in clinical development.

---

[1] Department of Urology, Zhongnan Hospital of Wuhan University, Wuhan 430071, China. [2] Frontier Science Center for Immunology and Metabolism, Medical Research Institute, Wuhan University, Wuhan 430071, China. [3] Department of Pathophysiology, School of Basic Medical Sciences, Wuhan University, Wuhan 430071, China. ✉email: qingguoliang@whu.edu.cn

The MYC family oncogene is deregulated in >50% of human cancers, frequently correlating with poor prognosis and unfavorable patient survival[1–4]. MYC family contains three members, MYC, MYCN, and MYCL, which encodes MYC (also called c-MYC), MYCN, and MYCL, respectively. MYC is the most broadly deregulated oncogene in a wide range of malignant tumors, whereas MYCN is frequently amplified in neural tumors such as neuroblastoma and small cell lung carcinoma, and MYCL amplifications are solely reported in small cell lung carcinoma[1,3].

The MYC oncoproteins are "super-transcription factors" that mediate a transcriptional response involved in a variety of biological processes, contributing to almost every aspect of tumorigenesis[2]. The wide implication of MYC family oncoproteins in both tumor initiation and maintenance suggests that therapeutic targeting MYC expression/activity should achieve a significant clinical efficacy[5,6]. However, efforts to directly disrupt MYC function with specific inhibitors have not been successful owing to its "undruggable" protein structure[7], reinforcing the demand for both improved understanding of MYC deregulation and alternative strategies to target MYC.

EZH2 is the main enzymatic subunit of the polycomb repressive complex 2 (PRC2), which functions together with embryonic ectoderm development (EED) and suppressor zeste 12 (SUZ12) as a histone methyltransferase to catalyze histone H3 lysine 27 trimethylation (H3K27me3) and repress gene transcription[8]. High levels of EZH2 often correlate with tumor stage and poor prognosis, while genetic deletion of EZH2 can block proliferation and survival in tumor cell lines and mouse models. Consequently, EZH2 is a potential therapeutic target and several inhibitors are in development for clinical usage[9–11]. Epigenetic silencing of tumor suppressor gene expression via H3K27me3 modification has been viewed as a common mechanism accounting for EZH2 oncogenic functions[8,12]. As such, efforts to therapeutically target EZH2 have generally focused on inhibition of its methyltransferase activity[8,12], although it remains undefined whether modulation of H3K27 trimethylation is the prominent mechanism whereby EZH2 promotes cancer.

Using neuroblastoma and small cell lung carcinoma as model systems, we herein identify a previously underappreciated crosstalk between EZH2 and MYC(N) in regulation of tumor formation. We therefore seek to investigate the regulatory mechanisms and involving biological relevance in the current study.

## Results

### EZH2 directly binds MYC and MYCN via the MYC homology box 2 domain.
In an effort to identify common regulators of MYC and MYCN, we performed immunoprecipitation analysis using the total cellular extracts from MYCN-amplified neuroblastoma Kelly cells and MYC-amplified small cell lung carcinoma NCI-H2171 cells, which exhibit very high levels of endogenous MYCN and MYC, respectively. Both MYC(N)-bound immunoprecipitates from these cells were subjected to liquid chromatography tandem mass spectrometry (LC-MS/MS) analysis. A total of 1561 potential interactors were identified in MYC-bound immunoprecipitates, and 1469 in MYCN-bound ones, among which 1059 proteins (~70% of overlap) were common MYC and MYCN interactors (Fig. 1a–c). Of note, 54% (796/1469) of the MYCN interactomes was identical to those from Eilers lab[13] (Supplementary Fig. 1a), and 41% (645/1561) and 31% (497/1561) of the MYC interactomes was identical to those from Penn[14] and Beli lab[15], respectively (Supplementary Fig. 1b). In addition, MYC(N) interactomes from Eilers, Penn, and Beli labs were only partially overlapped. All these data support that the full spectra of MYC(N) interactomes vary depending on both the approaches and the cell lines used for interactome isolation.

Conceivably, MYC(N) would associate with distinct proteins in different biological contexts.

Interestingly, both subunits of the PRC2 complex, EZH2 and EED, emerged as top hits in addition to previously validated MYC(N) common binding partners (MAX, NCL, ELF2, etc., Fig. 1d). Co-immunoprecipitation using Kelly and NCI-H2171 cell lysates showed that endogenous EZH2, EED, and SUZ12 were all present in MYCN and MYC immunoprecipitates (Fig. 1e and Supplementary Fig. 1d). In support of this observation, previous studies showed that MYCN formed a complex with EZH2 to drive the transcriptional repression of downstream targets[16,17]. However, whether interaction between MYC(N) and the PRC2 complex is direct and which PRC2 subunit(s) is involved in remains largely unknown. We first confirmed that interaction between MYC(N) and the PRC2 complex is independent of DNA or RNA within the cell lysates, as treatment of the cell lysates with a combination of DNase and RNase barely affected MYC(N) association with PRC2 subunits (Fig. 1e and Supplementary Fig. 1c, d). We then performed GST-pulldown assays with recombinant His-tagged MYCN and GST-tagged EZH2, EED, or SUZ12 proteins, and found that EZH2, but not EED and SUZ12, directly bound MYCN (Fig. 1f). We further identified that the MYC homology box 2 (MB2), but not MB1 and MB3, domain within MYC(N) and the chromodomain Y-like protein binding region (CDYL-BR) within EZH2 are responsible for this direct MYC(N) and EZH2 interaction (Fig. 1g–i and Supplementary Fig. 1e). Taken together, our results identified EZH2 as a direct, high-confidence MYC and MYCN common binding partner.

### EZH2 depletion inhibits MYC(N) expression and transcriptional activity.
MYCN amplification (usually leads to high MYCN expression) occurs in 20–25% of neuroblastomas overall and 40% of high-risk cases[18], while elevated MYC expression correlates with poor prognosis in MYCN-nonamplified neuroblastoma[19]. We wondered whether EZH2 affected MYC(N) expression given their direct protein interaction. To evaluate this, we performed EZH2 depletion in a number of MYCN-amplified and -nonamplified cell lines. Intriguingly, depletion of EZH2 expression by specific sgRNAs led to a marked decline in MYCN protein abundance in both MYCN-amplified Kelly, SK-N-BE2, and SMS-KAN cells and MYCN-nonamplified NBLS cells which express high levels of MYCN[20] (Fig. 2a). Since most MYCN-nonamplified neuroblastoma cells exhibited high MYC levels, we also inhibited EZH2 expression in SK-N-AS, SH-SY5Y, and SK-N-SH cells. As expected, EZH2 inhibition caused a significant depletion of MYC expression in all these cells (Fig. 2a and Supplementary Fig. 2a), but not in the MYCN-nonamplified SHEP cells lacking MYC and MYCN expression (Supplementary Fig. 2b). Moreover, in SK-N-SH cells which express both high levels of MYC and low levels of MYCN, EZH2 depletion decreased both MYC and MYCN expression (Supplementary Fig. 2a). Consistent with previous data[21], EZH2 depletion also reduced the protein levels of PRC2 subunit SUZ12 and EED (Supplementary Fig. 2c). Most likely, depletion of EZH2 caused SUZ12 and EED degradation. Importantly, EZH2 depletion consistently abrogated MYC(N) expression in both p53 wild type and mutant neuroblastoma cells (Fig. 2a), arguing that p53 is not involved in this event. Knockdown of EZH2 expression by two specific shRNAs similarly depleted MYCN expression (Supplementary Fig. 2d, e). Instead, depletion of EZH1, a related homolog of EZH2, exhibited an undetectable effect (Supplementary Fig. 2f), supporting the functional link between MYC(N) deregulation and EZH2 dependence.

The fact that EZH2 promotes MYCN (and MYC) expression prompted us to examine whether its inhibition affected MYCN

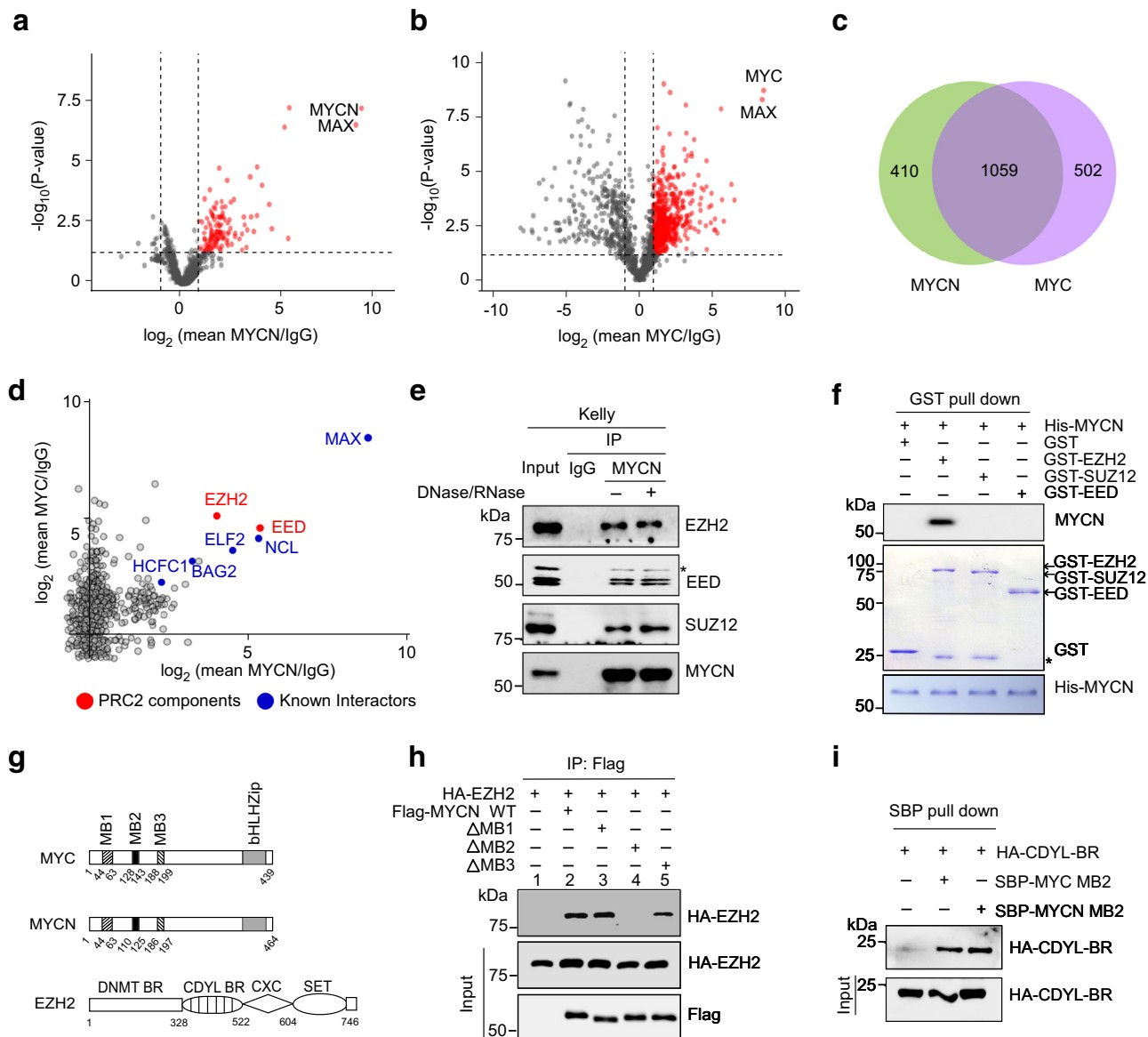

**Fig. 1 Identification of EZH2 as a MYCN/MYC common binding partner. a**, **b** Volcano plots showing interacting proteins of MYCN (**a**) and MYC (**b**). Associated proteins were immunoprecipitated from neuroblastoma Kelly cells (**a**) and small cell lung carcinoma NCI-H2171 cells (**b**) using specific MYCN and MYC antibodies, respectively. Proteins significantly enriched from three independent experiments with a fold-change >2 and *p* value < 0.05 against the IgG group are displayed as red dots. **c** Venn diagram of MYCN- and MYC-interacting partners. **d** Mass spectrometry results of MYCN and MYC common binding partners. The axes show the fold-change of α-MYCN/MYC-associated proteins relative to α-IgG control. **e** Co-immunoprecipitation (Co-IP) to detect interaction between endogenous MYCN and PRC2 components. Lysates of Kelly cells were treated with a DNase/RNase (10 μg/mL) combination and subjected to co-IP for detecting protein interaction. The asterisk denotes a nonspecific band. **f** GST pull-down to identify the direct MYCN-binding partner. Recombinant His-tagged MYCN was respectively incubated with GST-tagged EZH2, SUZ12, and EED in vitro before pulldown assay with GST beads, and MYCN interaction was analyzed by immunoblot with anti-His antibody. The asterisk denotes a nonspecific band. **g** Schematic representation of MYC, MYCN, and EZH2. MB1, 2, and 3: MYC box1, 2, and 3, respectively; bHLHZip: helix–loop–helix domain; DNMT BR: DNA methyltransferase-binding region; CDYL BR: chromodomain Y-like protein-binding region; CXC: cysteine-rich domain, SET: Su(var)3–9, enhancer of zeste, trithorax domain. **h** Characterization of the MB motif required for MYCN and EZH2 interaction. 293T cells were transfected with plasmids expressing epitope-tagged EZH2, MYCN, or its MYC box deletion (ΔMB) mutants as shown. Anti-Flag-associated precipitates were used for detection of EZH2 binding. **i** Analysis of EZH2 CDYL-BR and MYCN/MYC MB2 interaction. 293T cells were transfected with plasmids expressing EZH2-CDYL-BR, and Streptavidin-binding peptide (SBP)-tagged MYC or MYCN for 48 h as shown. SBP pull-down was performed to analyze the protein interaction as indicated. The experiments were independently repeated three times with similar results (**e**, **f**, **h**–**i**). Source data are provided as a Source data file.

transcriptional activities. To investigate this, we performed chromatin immunoprecipitation (ChIP) coupled to high-throughput sequencing (ChIP-seq) in *MYCN*-amplified SK-N-BE2 cells with antibodies recognizing MYCN and RNA polymerase II (pol II). ChIP-seq analysis demonstrated that MYCN loss upon EZH2 depletion markedly decreased MYCN

genome-wide occupancy while RNA polymerase II occupancy was minimally affected (Fig. 2b). Indeed, at the exemplary MYCN target genes *NCL* and *PRMT5*, EZH2 depletion compromised MYCN occupancy (Fig. 2c), which was further confirmed by ChIP assay in independent experiments (Fig. 2d). We also analyzed multiple representative MYCN targets in SK-N-BE2

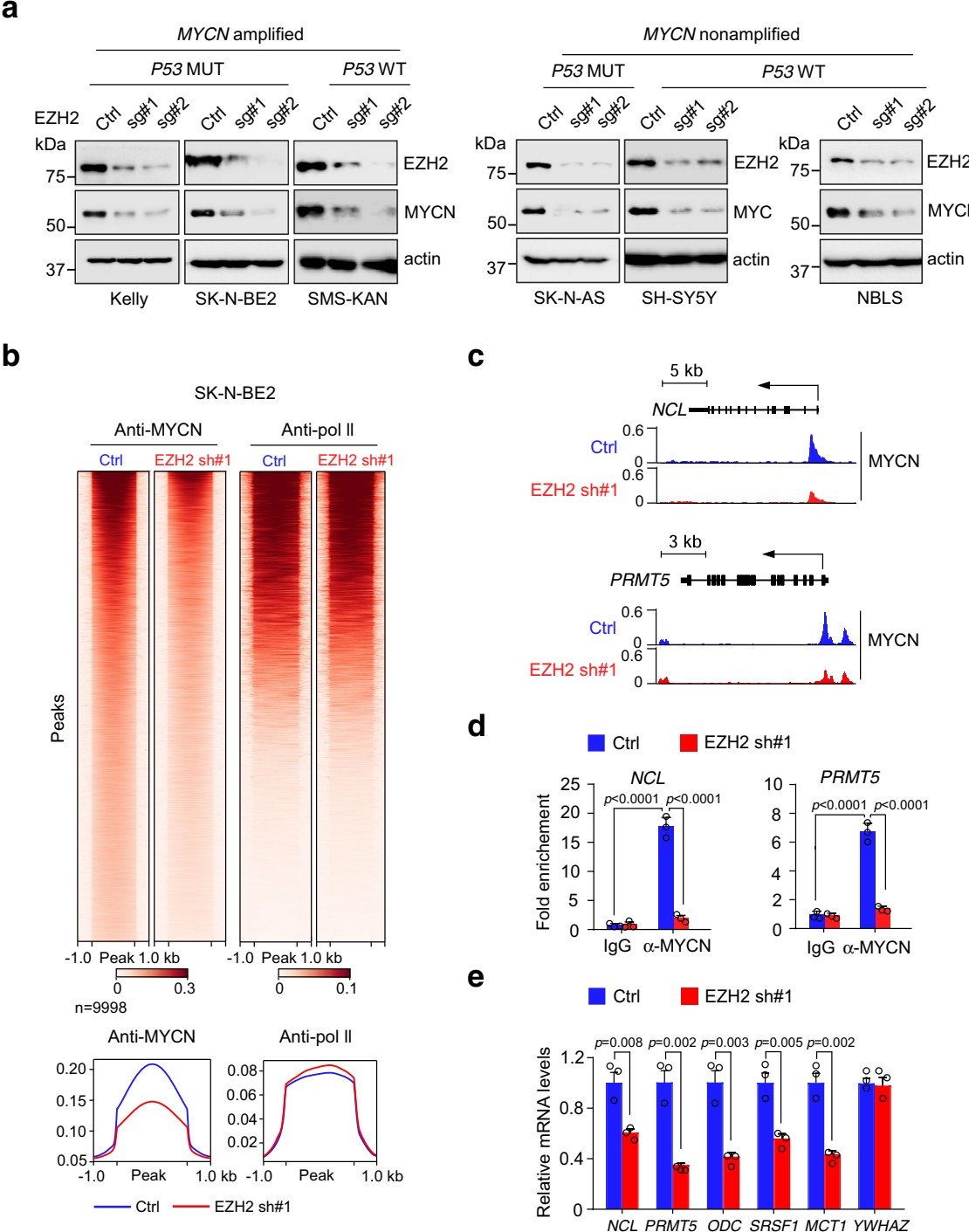

**Fig. 2 EZH2 deficiency decreases MYCN and MYC expression and transcriptional activity in neuroblastoma cells. a** Immunoblot to detect MYCN or MYC levels upon EZH2 depletion with specific sgRNAs in *MYCN*-amplified (Kelly, SK-N-BE2 and SMS-KAN) and *MYCN*-nonamplified (SK-N-AS, SH-SY5Y and NBLS) neuroblastoma cells. Actin was used as a loading control. The experiments were independently repeated three times with similar results. **b** Heatmap presentation of genome-wide MYCN and pol II occupancies in *MYCN*-amplifed SK-N-BE2 cells analyzed by ChIP-Seq. **c** Genome browser views of MYCN occupancy in the *NCL* and *PRMT5* loci in SK-N-BE2 cells with or without EZH2 depletion. **d** Binding of MYCN to the *NCL* or *PRMT5* promoter was analyzed by ChIP-qPCR in SK-N-BE2 cells with or without EZH2 depletion. Averages of fold enrichment between anti-MYCN and the isotype IgG control are shown. Graph shows mean ± SD from triplicates; significance was determined by two-way ANOVA test followed by Tukey's correction. **e** Real-time qPCR analysis of representative MYCN target genes in SK-N-BE2 cells upon EZH2 depletion. Graph shows mean ± SD from triplicates; significance was determined by unpaired two-tailed Student's *t*-test. Source data are provided as a Source data file.

cells by real-time qPCR analysis. *YWHAZ*, which is not regulated by MYCN, was used as a negative control. As expected, EZH2 knockdown significantly decreased these MYCN target gene expression (Fig. 2e), supporting that EZH2 was essential for MYCN transcriptional activities.

**EZH2 antagonizes FBW7α-mediated MYCN degradation**. We then sought to understand how EZH2 regulates MYCN expression. Depletion of EZH2 expression in multiple *MYCN*-amplified and -nonamplified neuroblastoma cells by specific sgRNAs or shRNAs caused minimal changes in *MYCN* mRNA levels (Supplementary Figs. 2d and 3a). Of note, administration of MG132 (a specific 26S proteasome inhibitor) almost completely rescued EZH2-depletion induced MYCN loss (Fig. 3a and Supplementary Fig. 3b), arguing that EZH2 promotes MYCN stabilization. We then performed time-course analysis and found that sgRNA depletion of EZH2 expression significantly shortened the half-life of endogenous MYCN in both Kelly and NBLS cells (Fig. 3b and Supplementary Fig. 3c). Similar results were obtained in MYC-driven SK-N-AS cells, supporting that EZH2 globally sustains MYC(N) protein stability in neuroblastoma cells.

Initiation of MYC(N) degradation involves phosphorylation of Threonine-58 (T58) within MYC and Threonine-50 (T50) within MYCN by glycogen synthase kinase 3β (GSK3β), respectively. Because the antibody against MYCN T50 was not commercially available, we chose to examine MYC T58 phosphorylation as a substitute given both members share the same phosphorylation event. Ectopic expression of either wild-type GSK3β or the S9A kinase-active mutant caused enhanced MYC T58 phosphorylation, while ectopically expressed EZH2 exhibited minimal effects on this phosphorylation (Supplementary Fig. 3d). In support of this observation, T50A phosphorylation-inactive mutant (abolished GSK3β-mediated phosphorylation of MYCN) bound EZH2 as capable as the wild-type MYCN (Supplementary Fig. 3e), and deletion of the whole MB1 (harbors the MYCN phosphor-degron) within MYCN had an undetectable effect on MYCN–EZH2 interaction (Fig. 1h, compare lane 3 vs lane 2). These results suggest that stabilization of MYC(N) by EZH2 occurs via a phosphorylation-independent mechanisms.

In contrast, depletion of EZH2 in Kelly cells resulted in a marked increase in polyubiquitination of endogenous MYCN (Fig. 3c). Previous studies showed that the ubiquitin E3 ligase FBW7 (ref. [22]) and HUWE1 (ref. [23]) predominantly regulate proteasome-mediated degradation of MYC family oncoproteins. Of note, knockdown of FBW7α in Kelly cells rescued EZH2-depletion induced MYCN loss, whereas knockdown of HUWE1 exhibited little effect (Fig. 3d). As such, shRNA-mediated knockdown of FBW7α elicited a time-dependent rescue of MYCN loss resulting from EZH2 abrogation (Supplementary Fig. 3f). Moreover, expression of increasing amounts of EZH2 dramatically inhibited MYCN polyubiquitination by FBW7α (Fig. 3e, left panel: compare lane 2 vs lanes 3–5), leading to a dose-dependent effect on MYCN stabilization (Fig. 3f, compare lane 2 vs lanes 4–6). Consistently, ectopic expression of EZH2 elicited a dramatic increase in MYCN half-life upon co-transfection of FBW7α in 293T cells (Fig. 3g), further supporting that EZH2 promotes MYCN stabilization via abrogation of FBW7α function.

**EZH2 regulates MYCN stabilization independent of its methyltransferase activity**. In addition to function as a primary histone H3K27 methyltransferase, EZH2 also regulates the methylation of nonhistone substrates[24,25]. The implication of EZH2 in direct methylation of nonhistone proteins prompted us to explore whether it promotes MYCN stabilization in a methyltransferase-dependent manner. To this end, we performed

in vitro methylation assays and found that purified PRC2 core complex efficiently methylated the histone octamer (Supplementary Fig. 4a, lane 2), but not the recombinant His-tagged MYCN (Supplementary Fig. 4a, lane 5). Note that the PRC2 core complex containing methyltransferase-inactive EZH2 (H689Y)[26] failed to methylate the histone octamer (Supplementary Fig. 4a, compare lane 2 vs lane 3). A methylated protein band above 75 kDa of molecular weight was observed in the assay, indicating an automethylation of SUZ12 by PRC2 (ref. [27]). These results support that regulation of MYCN stabilization by EZH2 occurs via mechanisms independent of direct MYCN methylation.

It was then expected that the methyltransferase-inactive EZH2 should, in principle, stabilize MYCN as efficiently as the wild-type (WT) form. Indeed, co-expression of the EZH2 methyltransferase-inactive EZH2-H689A mutant, which is capable of binding with MYCN (Supplementary Fig. 4b), similarly inhibited MYCN polyubiquitination and degradation by FBW7α (Fig. 4a, b and Supplementary Fig. 4c). Expression of increasing amounts of EZH2-H689A also dose-dependently inhibited FBW7α-mediated MYC degradation.

We next examined whether stabilization of endogenous MYCN is primarily methyltransferase-independent. Kelly cells were treated with a specific shRNA targeting *EZH2* 3′-UTR to deplete endogenous EZH2, which was then subjected to rescue using WT or H689A-mutant EZH2. As expected, MYCN protein levels were significantly decreased upon knockdown of endogenous EZH2 in comparison to mock control (Fig. 4c, compare lane 1 vs lane 4), and were fully rescued upon re-introduction of either WT or H689A-mutant EZH2 (Fig. 4c, compare lane 4 vs lanes 5–6). By contrast, the decline in H3K27me3 due to knockdown of endogenous EZH2 can only be rescued by WT EZH2 but not the methyltransferase-inactive mutant (Fig. 4c, compare lanes 4 and 6 vs lane 5). We then treated Kelly cells with distinct EZH2 inhibitors, DZNep (3-deazaneplanocin A), GSK126, or EPZ6438. Consistent with a previous study[21], DZNep depleted EZH2 protein as a whole and reduced the levels of SUZ12 and EED as well (Fig. 4d and Supplementary Fig. 4d), while GSK126 and EPZ6438 inhibited EZH2 methyltransferase activity without affecting the protein levels of EZH2 and other PRC2 components (Fig. 4d and Supplementary Fig. 4d). Again, administration of DZNep markedly decreased MYCN protein levels in Kelly cells (Fig. 4d), while treatment of cells with GSK126 or EPZ6438 exhibited minimal effects on MYCN (and EZH2) expression although leading to a similar decrease in H3K27me3 levels (Fig. 4d). Addition of MG132 rescued DZNep-induced MYCN loss in Kelly cells (Supplementary Fig. 4e). Moreover, treatment with DZNep accelerated MYCN turnover, which was pronounced upon addition of CHX to block the de novo protein synthesis (Supplementary Fig. 4f). All these results argue that stabilization of MYCN is independent of EZH2 methyltransferase activity.

Previous data showed that Aurora kinase A (AURKA) interacted with both MYCN and FBW7α, and counteracted FBW7α-mediated MYCN degradation in neuroblastoma cells[28,29]. It is possible that EZH2 forms a joint complex with AURKA and stabilizes MYCN via an AURKA-dependent mechanism. However, immunoprecipitation assays revealed that EZH2 was absent in the AURKA immunoprecipitates when both proteins were co-expressed in 293T cells, while considerable amounts of EZH2 were present in the MYCN immunoprecipitates at the same conditions (Supplementary Fig. 4g, compare lane 3 vs lane 2). Moreover, endogenous AURKA and EZH2 failed to form a detectable complex in vivo (Supplementary Fig. 4h), and ectopic expression of AURKA did not enhance the interaction between EZH2 and MYCN (Supplementary Fig. 4i). All these data argue that EZH2 counteracted FBW7α-mediated MYCN degradation largely independent of AURKA.

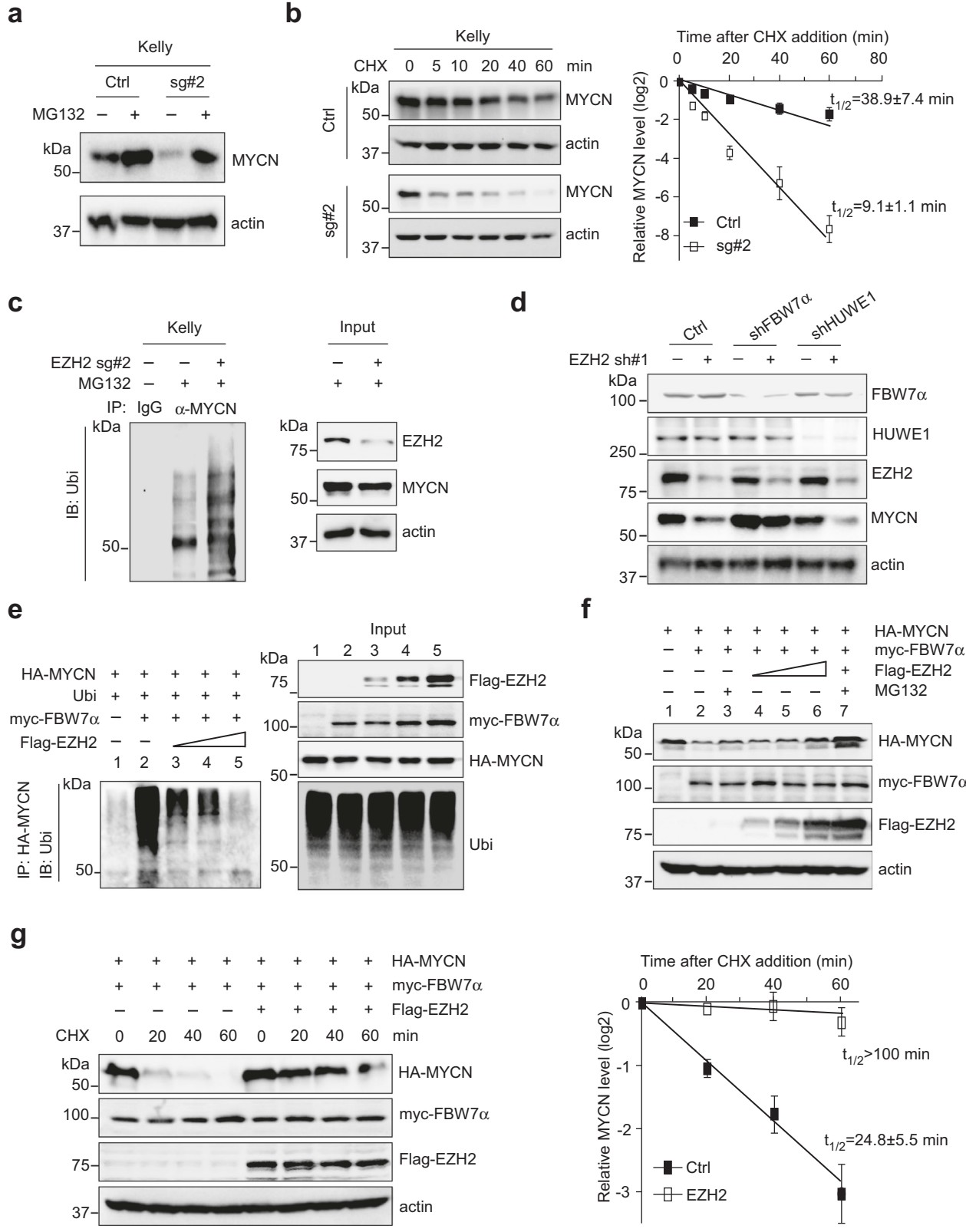

Since EZH2 directly interacted with MYCN, but not FBW7α, we considered the possibility that elevated levels of EZH2 compete with FBW7α for access to MYCN. We therefore examined whether increasing amounts of EZH2 displaced FBW7α from binding to MYCN. We used the F-box deletion mutant of FBW7α (FBW7αΔF) as a substitute because WT FBW7α immediately targeted MYCN for proteasomal

degradation. As expected, the EZH2 methyltransferase-inactive mutant (EZH2 H689A) was as capable as the WT EZH2 in dose-dependently displacing MYCN from a complex with FBW7α when all three proteins were co-expressed by transient transfection in 293T cells (Fig. 4e, compare lanes 3–5 vs lane 2, and lanes 6–8 vs lane 2). When reconstituted with recombinant proteins in vitro, EZH2 also similarly displaced FBW7α from binding to

**Fig. 3 EZH2 stabilizes MYCN protein by counteracting FBW7α-mediated degradation. a** Immunoblots of MYCN in Kelly cells with or without EZH2 depletion. Cells were treated with MG132 (5 μM) for 6 h before harvest as indicated. **b** Time-course analysis of MYCN protein levels in EZH2-depleted Kelly cells from one representative experiment (left). MYCN band density relative to actin was quantified and plotted on the right. Data shown were obtained from averages of three independent experiments. **c** Polyubiquitination of MYCN was analyzed in EZH2-depleted Kelly cells. Kelly cells with or without EZH2 depletion were treated with MG132 (5 μM) for 6 h as indicated. Ubiquitin-conjugated MYCN proteins from Kelly lysates were immunoprecipitated by anti-MYCN antibody and subjected to immunoblot with ubiquitin antibody. **d** Analysis of MYCN expression in Kelly cells expressing a specific shRNA targeting FBW7α or HUWE1. Kelly cells were infected with indicated shRNAs for 48 h. Immunoblots of indicated proteins are shown. **e, f** Analysis of MYCN polyubiquitination and protein abundance with increasing doses of EZH2. 293T cells were transfected with plasmids expressing epitope-tagged MYCN, ubiquitin, FBW7α and/or increasing amounts of EZH2 as indicated. Ubiquitin-conjugated MYCN proteins were immunoprecipitated with HA-tag antibody and subjected to immunoblot with ubiquitin antibody (**e**). 293T cells were transfected with plasmids expressing MYCN, FBW7α, and increasing amounts of EZH2 for 48 h. MYCN and other indicated proteins were analyzed by immunoblot (**f**). **g** Time-course analysis of HA-MYCN levels in 293T cells expressing ectopic HA-MYCN, myc-FBW7α, and/or Flag-EZH2 as indicated from one representative experiment (left). MYCN band density relative to actin was quantified and plotted on the right. Data shown were obtained from averages of three independent experiments. The experiments were independently repeated three times with similar results (**a**, **c–f**). Source data are provided as a Source data file.

MYCN in a dose-dependent manner (Fig. 4f, compare lanes 4–6 vs lane 3), arguing that EZH2 sustained MYC stabilization by competing against FBW7α in a methyltransferase-independent manner. Most likely, direct binding of EZH2 to MB2 would cause spacial hindrance and interfere with recognition of MB1 by FBW7α given MB1 and MB2 are closely located within the N-terminus of MYCN.

**Depletion of EZH2, but not enzymatic inhibition, inhibits neuroblastoma tumor cell growth in a MYC(N)-dependent manner.** The fact that the methyltransferase-inactive EZH2 is capable of modulating MYC(N) stability prompted us to examine its functional relevance in MYC(N)-driven neuroblastoma cells. We respectively generated Kelly cells expressing the control vector, EZH2 WT or H689A mutant. These cells were then treated with a control or specific shRNA targeting EZH2 3′-UTR to knockdown endogenous EZH2. As expected, depletion of endogenous EZH2 almost completely inhibited Kelly cell proliferation, whereas re-introduction of H689A mutant, which was as capable as EZH2 WT, significantly rescued cell proliferation (Fig. 5a). Ectopic expression of EZH2 WT or H689A in control shRNA treated Kelly cells had minimal effect on cell proliferation, arguing that the already-high levels of endogenous EZH2 was sufficient to sustain its oncogenic functions. In contrast, ectopic expression of SUZ12 (another PRC2 subunit) failed to rescue MYCN expression and cell proliferation resulting from knockdown of endogenous EZH2 (Supplementary Fig. 5a). These data highlight a specific role of EZH2 in MYCN stabilization and MYCN-driven neuroblastoma cell growth.

We then used pharmacologic inhibitors to recapitulate our observation in MYCN-driven neuroblastoma Kelly and BE-2C cells. As expected, administration of DZNep, which depleted EZH2 and MYCN, markedly inhibited proliferation of all the tumor cells examined, while administration of the enzymatic inhibitors (GSK126 and EPZ6438) exhibited minimal effects on Kelly and BE-2C cells (Fig. 5b and Supplementary Fig. 5b). DZNep is an inhibitor of S-adenosyl-L-homocysteine hydrolase[30], in principle, it could also target additional S-adenosyl-L-methionine-dependent methyltransferases besides EZH2. However, administration of DZNep failed to exhibit further inhibitory effects on MYCN expression and cell proliferation when endogenous EZH2 was depleted (Supplementary Fig. 5c), suggesting that EZH2 is a primary DZNep target in the current cell contexts. Moreover, ectopic expression of MYCN T50A mutant, which abolished GSK3β-mediated phosphorylation and circumvented MYCN degradation by proteasome, significantly reversed growth inhibition by either knockdown of endogenous EZH2 or administration of DZNep (Fig. 5c, d). Similar results were

obtained in MYC-driven SK-N-AS cells, where ectopic expression of the MYC T58A mutant also significantly reversed growth inhibition upon EZH2 knockdown (Supplementary Fig. 5d). These results support that methyltransferase-independent regulation of MYC(N) stabilization is essential for EZH2 oncogenic functions.

We next established subcutaneous xenografts using MYCN-amplified Kelly cells to confirm whether these EZH2 inhibitors could affect the sensitivity of established tumors in a MYCN-dependent manner. Mice with palpable tumors were randomized into nine groups, each receiving a different treatment regimen. Consistent with the in vitro findings, administration of DZNep effectively depleted EZH2 and MYCN in vivo, and caused a significant, dose-dependent inhibition of Kelly tumors, whereas these tumors were found to be highly resistant to GSK126 treatment even though a greater H3K27me3 inhibition was achieved (Fig. 5e, f). When the current work was undergoing, a study showed that administration of 150 mg/kg of GSK126 significantly inhibited the Kelly xenograft tumor growth[31]. Similar effect was achieved in the current study when GSK126 was used at the same high dose (Fig. 5e). However, at this dose, both EZH2 and MYCN were significantly diminished (Fig. 5f), which might explain (at least partially) why only very high doses of GSK126 achieved considerable therapeutic efficacy. Actually, administration of GSK126 at the dose of 25 mg/kg already exhibited significant in vivo efficacy in terms of H3K27me3 depletion (Fig. 5f, compare lane 5 vs lanes 1 and 2), yet the MYCN abundance and xenograft tumor growth was largely unchanged at this dose (Fig. 5e), suggesting that MYCN stabilization is more critical for EZH2 oncogenic function in the current tumor contexts. All of these findings provide strong evidence demonstrating that a non-catalytic role of EZH2 in MYCN stabilization and tumor progression in vivo, arguing that the enzymatic inhibitors now in clinical development may be impotent in MYC(N)-driven neuroblastomas unless they are capable of depleting EZH2 and/or MYC(N).

**Ezh2 conditional knockout effectively depletes MYCN and impedes MYCN-driven neuroblastoma.** The differential efficacy of EZH2 inhibitors in neuroblastoma tumor cell lines prompted us to further evaluate Ezh2 regulation of MYCN in mice. We first performed Ezh2 knockdown in primary neuroblastoma cells isolated from the TH-MYCN transgenic mice[32]. Consistently, knockdown of mouse Ezh2 by specific shRNAs led to a marked decrease in both MYCN protein levels and primary neuroblastoma cell growth (Fig. 6a). Again, Ezh2 depletion exhibited minimal effects on MYCN mRNA levels (Supplementary Fig. 6a). As expected, administration of DZNep depleted MYCN

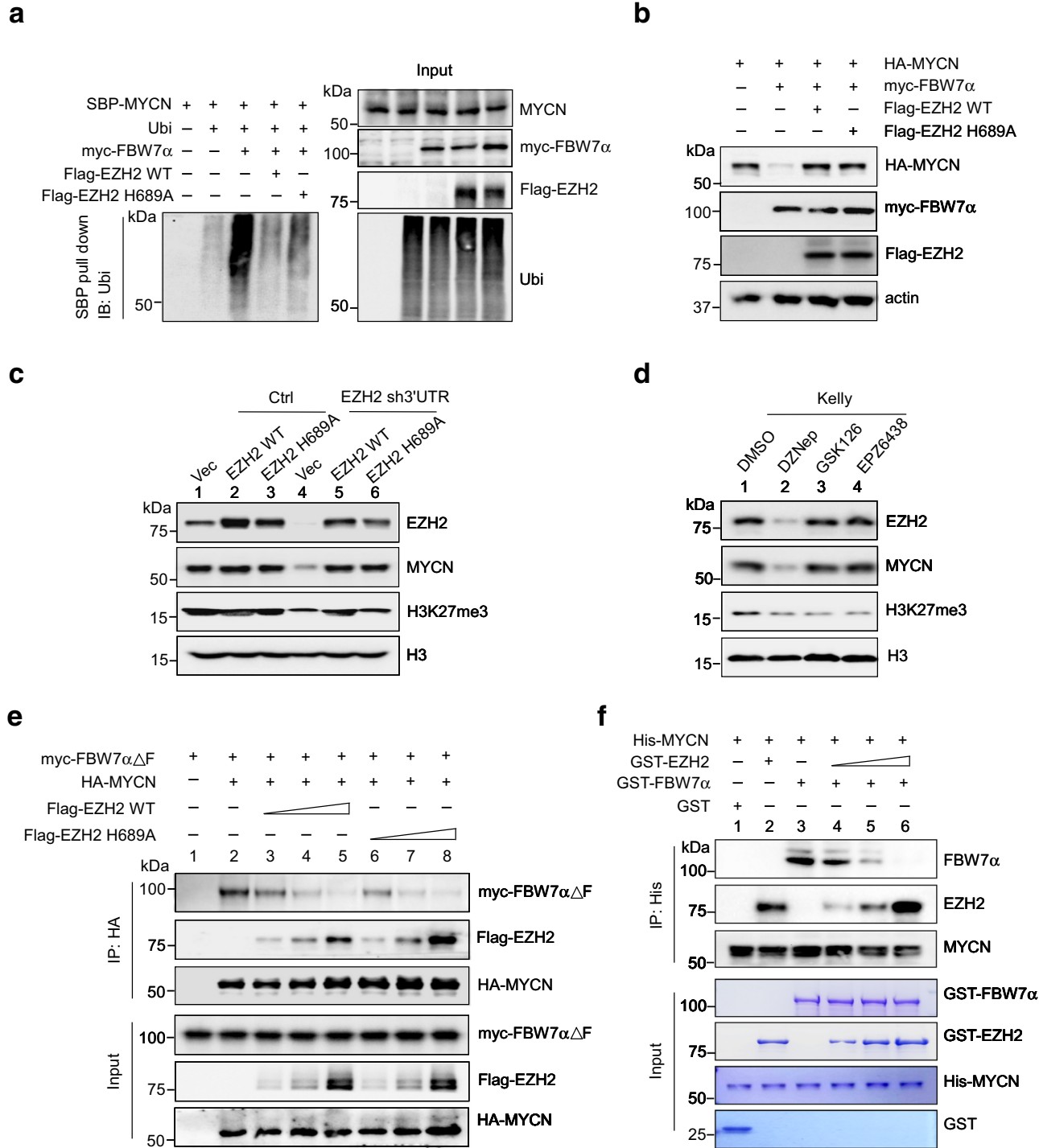

(and Ezh2) concomitantly with a dramatic inhibition of primary tumor cell growth in vitro, whereas the enzymatic inhibitor GSK126 exhibited minimal effects (Fig. 6b).

We then established neuroblastoma allografts in syngeneic 129X1/SvJ mice using tumor sections from the TH-*MYCN* transgenic mice. Based on the Kelly xenograft experiments, we respectively used 3 mg/kg of DZNep as well as 50 and 100 mg/kg of GSK126 for TH-*MYCN* allograft treatment. Again, administration of 3 mg/kg of DZNep effectively inhibited the tumor growth, whereas these tumors were found to be similarly resistant to both 50 and 100 mg/kg of GSK126 treatment (Supplementary Fig. 6b). We then tested 3 mg/kg of DZNep and 50 mg/kg of GSK126 in

TH-*MYCN* mice. Homozygous TH-*MYCN* mice were treated with DZNep, GSK126, or vehicle control at the time a palpable tumor was documented. As a result, 3 mg/kg of DZNep therapy markedly prolonged survival of tumor-bearing mice in comparison with either 50 mg/kg of GSK126 or mock treatment (Fig. 6c). More importantly, only tumors treated with DZNep exhibited MYCN (and EZH2) loss even though 50 mg/kg of GSK126 was as capable as 3 mg/kg of DZNep in diminishing H3K27me3 expression within tumor cells (Fig. 6d, compare the H3K27me3 immunochemistry images between GSK126 and DZNep treatment).

We next initiated a neural crest-specific deletion of *Ezh2* by mating TH-*MYCN* mice with *Ezh2*^f/f^(flox/flox) mice and then

**Fig. 4 EZH2 regulates MYCN stabilization independent of its methyltransferase activity. a** Analysis of MYCN polyubiquitination in the presence of WT EZH2 or the methyltransferase-inactive mutant (H689A). 293T cells were transfected with plasmids expressing epitope-tagged MYCN, ubiquitin, FBW7α and WT or H689A EZH2 for 48 h as shown. Ubiquitin-conjugated MYCN proteins were pulled down with SBP beads and subjected to immunoblot with ubiquitin antibody. **b** Comparison of MYCN protein levels in 293T cells expressing EZH2 WT or H689A. 293T cells were transfected with plasmids expressing epitope-tagged MYCN, FBW7α, and EZH2 (WT or H689A) for 48 h as shown. MYCN and other indicated proteins were analyzed by immunoblots. **c** Kelly cells upon shRNA depletion of endogenous EZH2 (targeting *EZH2* 3′UTR) were transduced with EZH2 WT or H689A. MYCN and other indicated proteins were analyzed by immunoblots. Histone H3 was used as a loading control. **d** Immunoblot of MYCN in the presence of various EZH2 inhibitors. Kelly cells were respectively treated with 5 μM of DZNep, GSK126, or EPZ6438 for 48 h before subjected to immunoblotting. **e** Co-immunoprecipitation (co-IP) to detect the protein-protein between MYCN and FBW7αΔF in the presence of EZH2 WT and H689A. 293T cells were transfected with plasmids expressing epitope-tagged MYCN, FBW7αΔF (F-box deletion mutant) and increasing amounts of EZH2 WT or H689A mutant for 48 h as shown. HA-MYCN-associated proteins were analyzed by immunoblot. **f** In vitro pull-down experiments to detect the protein–protein between MYCN and EZH2 (or FBW7α). Recombinant His-tagged MYCN was incubated with GST-tagged FBW7α and/or increasing amounts of EZH2. MYCN-associated proteins were precipitated by nickel beads and analyzed by immunoblotting. The experiments were independently repeated three times with similar results (**a**–**f**). Source data are provided as a Source data file.

with TH-cre^ERT2 mice[33,34] (Supplementary Fig. 6c, d). Administration of tamoxifen (TAM) results in creER recombinase-mediated *Ezh2* deletion in peripheral neural crest cells that give rise to neuroblastoma. We then isolated primary sympathetic ganglia from *Ezh2*-deleted TH-*MYCN* mice and age-matched controls (Supplementary Fig. 6d). As expected, TAM induction of *Ezh2* deletion depleted MYCN, significantly inhibited expression of representative MYCN metabolic targets and uptake of the radiotracer ^18F-deoxyglucose (FDG) to tumor sites (Fig. 6e, f), and dramatically prolonged the survival of TH-*MYCN* mice (Supplementary Fig. 6e). Altogether, these results support that Ezh2 is indispensable for sustaining MYCN deregulation and maintaining MYCN-driven neuroblastoma in vivo.

We further confirmed EZH2 and MYCN protein levels by immunoblot in 18 primary tumor samples isolated from TH-*MYCN* mice. Of note, expression of EZH2 and MYCN was significantly correlated, while the wild-type ganglia barely expressed both proteins (Fig. 6g, h). All these data argue that the EZH2-MYCN axis sustains MYCN expression and amplifies MYCN-dependent oncogenic programs to promote neuroblastoma progression.

**EZH2 depletion induces MYC(N) degradation in small cell carcinoma and Burkitt's lymphoma cells and profoundly inhibits xenograft tumor growth.** As stated above, we have shown that EZH2 plays an essential role in MYC(N) stabilization in neuroblastoma cells. These findings prompted us to examine whether similar regulation occurs in other MYC(N) driven tumor cells. To evaluate this, we performed EZH2 depletion in representative MYC(N)-driven small cell carcinoma NCI-H526 and NCI-H82 cells as well as MYC-driven Burkitt's lymphoma Daudi cells, which are characterized by MYC(N) overexpression due to chromosome amplification or translocation. As expected, depletion of EZH2 expression by specific sgRNAs significantly reduced MYC(N) protein abundance in all these cells with minimal effect on mRNA levels (Fig. 7a–c and Supplementary Fig. 7a). Addition of MG132 consistently rescued the MYCN loss caused by EZH2 depletion in NCI-H526 cells (Supplementary Fig. 7b, upper panel), arguing that EZH2 regulates MYCN stabilization in these tumor cells. As such, time-course analysis revealed that inhibition of EZH2 expression significantly shortened the half-life of endogenous MYCN (Supplementary Fig. 7c). Addition of MG132 similarly rescued the MYC loss caused by EZH2 depletion in NCI-H82 cells (Supplementary Fig. 7b, lower panel), and ectopic EZH2 expression significantly extended the half-life of MYC when both MYC and FBW7α were co-expressed in 293T cells (Supplementary Fig. 7d).

Again, administration of DZNep, which depleted EZH2 and MYC(N) (Fig. 7a–c), markedly inhibited proliferation of all the

tumor cells examined, while administration of the enzymatic inhibitor GSK126 largely exhibited minimal effect (Fig. 7a–c). Addition of 5 μM of GSK126 in NCI-H82 cells, to some degree, reduced MYC expression and subsequently inhibited cell proliferation (Fig. 7b). It is possible that a dose of 5 μM GSK126 partially disrupted EZH2-MYC interaction, leading to a portion of MYC degradation in NCI-H82 cells. We then established subcutaneous xenografts using NCI-H526 and NCI-H82 cells. Mice with palpable tumors were randomized into five groups, each receiving a different treatment regimen. Consistent with the in vitro findings, administration of DZNep effectively depleted EZH2 and MYC(N) in vivo (Supplementary Fig. 7e, f), and caused a significant, dose-dependent inhibition of both NCI-H526 and NCI-H82 tumors (Fig. 7d, e). Treatment of NCI-H526 xenografts with a dose of either 50 or 100 mg/kg of GSK126 barely affected MYCN expression and tumor growth (Fig. 7d). Yet, administration of the same dose of GSK126 caused MYC depletion and tumor growth inhibition in NCI-H82 xenografts (Fig. 7e and Supplementary Fig. 7e, f), indicating that GSK126 at these doses may disrupt EZH2-MYC interaction and induce MYC degradation in vivo. Taken together, these data suggest that EZH2 depletion induces MYC(N) degradation not only in neuroblastoma cells but also in other cancer cells overexpressing either the *MYCN* or *MYC* oncogene.

## Discussion

Epigenetic silencing of tumor suppressor gene expression via H3K27me3 modification has been viewed as a predominant mechanism accounting for EZH2 oncogenic functions[12]. In the current study, we identified EZH2, independent of its methyltransferase activity, as an essential regulator of MYC(N) stabilization and the stabilized MYC(N) proteins in turn as essential EZH2 oncogenic effectors. These findings decipher a previously unsuspected mechanism involved in MYC(N) deregulation, and establish MYC family super-transcription factors as universal mediators in sustaining EZH2 oncogenic programs given that deregulation of MYC(N) occurs in >50% of human cancers.

EZH2 is overexpressed in numerous tumor entities, including neuroblastoma, small cell carcinoma, and prostate cancer, and is frequently associated with aggressive disease, leading to its classification as an oncogene[12,35]. As the major H3K27me3 methyltransferase, EZH2 is commonly believed to execute its tumor-promoting function via transcriptional repression of tumor suppressor genes. Paradoxically, we identify a critical function of EZH2 in human cancers is to stabilize MYC-family oncoproteins and sustain MYC(N) dependent transcriptional amplification independent of its methyltransferase activity. Most likely, both mechanisms would act in concert to promote tumor initiation and progression.

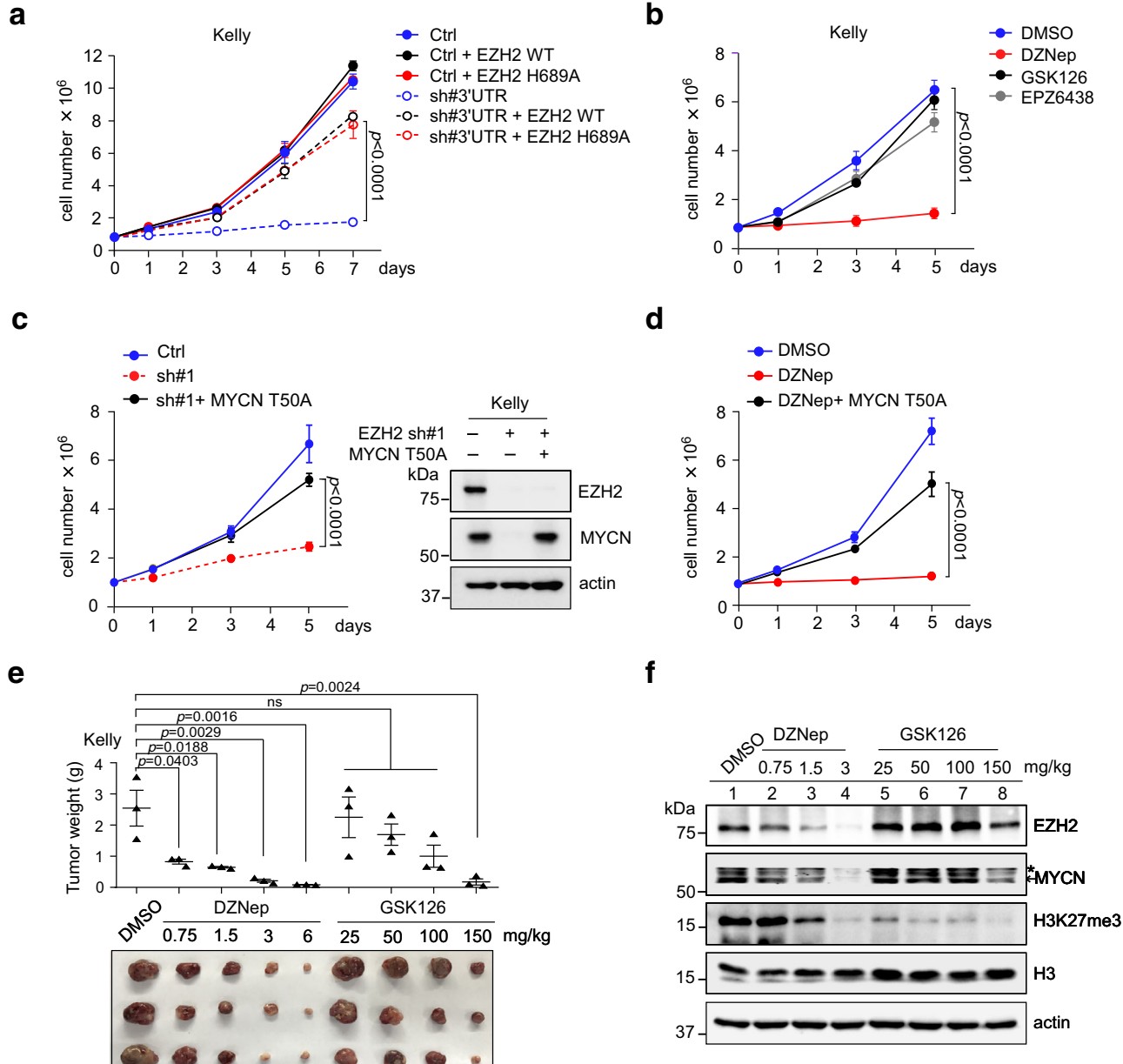

**Fig. 5 Depletion of EZH2, but not enzymatic inhibition, inhibits tumor cell growth in a MYCN-dependent manner. a** Proliferation of Kelly cells was measured by serial cell counts upon shRNA depletion of endogenous EZH2 in the presence or absence of ectopically expressed EZH2 WT or H689A. Data shown were obtained from averages (±SD) of technical triplicates; significance was determined by two-way ANOVA test followed by Tukey's correction. **b** Proliferation of Kelly cells was measured upon administration of 5 μM of DZNep, GSK126, or EPZ6438. Data shown were obtained from averages (±SD) of technical triplicates; significance was determined by two-way ANOVA test followed by Tukey's correction. **c** Proliferation of Kelly cells was measured upon shRNA depletion of endogenous EZH2 with or without ectopic expression of the MYCN T50A. Data shown were obtained from averages (±SD) of technical triplicates; significance was determined by two-way ANOVA test followed by Tukey's correction. **d** Proliferation of Kelly cells was measured upon DZNep treatment (5 μM) with or without ectopic expression of the MYCN T50A. Data shown were obtained from averages (±SD) of technical triplicates; significance was determined by two-way ANOVA test followed by Tukey's correction. **e** Kelly xenografts were subjected to DZNep or GSK126 treatments at the indicated doses. Images of subcutaneous tumors (n = 3 per group) under various treatments are shown. Graph shows mean ± SD from biological triplicates; significance was determined by one-way ANOVA test followed by Tukey's correction. **f** Immunoblots of indicated proteins in Kelly xenograft tumors. The experiments were independently repeated three times with similar results. Source data are provided as a Source data file.

EZH2 holds great promise as a therapeutic target. As such, multiple EZH2 inhibitors, which inhibit the H3K27me3 methyltransferase activity (that is, reducing H3K27me3) of EZH2/PRC2, have entered numerous clinical trials[35]. However, trials performed to date demonstrate that these inhibitors exhibit quite differential clinical efficacies, arguing that selection of

proper patient populations is a critical determining factor. In this regard, US Food and Drug Administration (FDA) has recently approved Tazemetostat (EPZ6438) for the treatment of patients with relapsed/refractory follicular lymphomas harboring EZH2-activating mutations (thus more dependent on EZH2 enzymatic activity)[36]. However, EZH2-activating mutations are

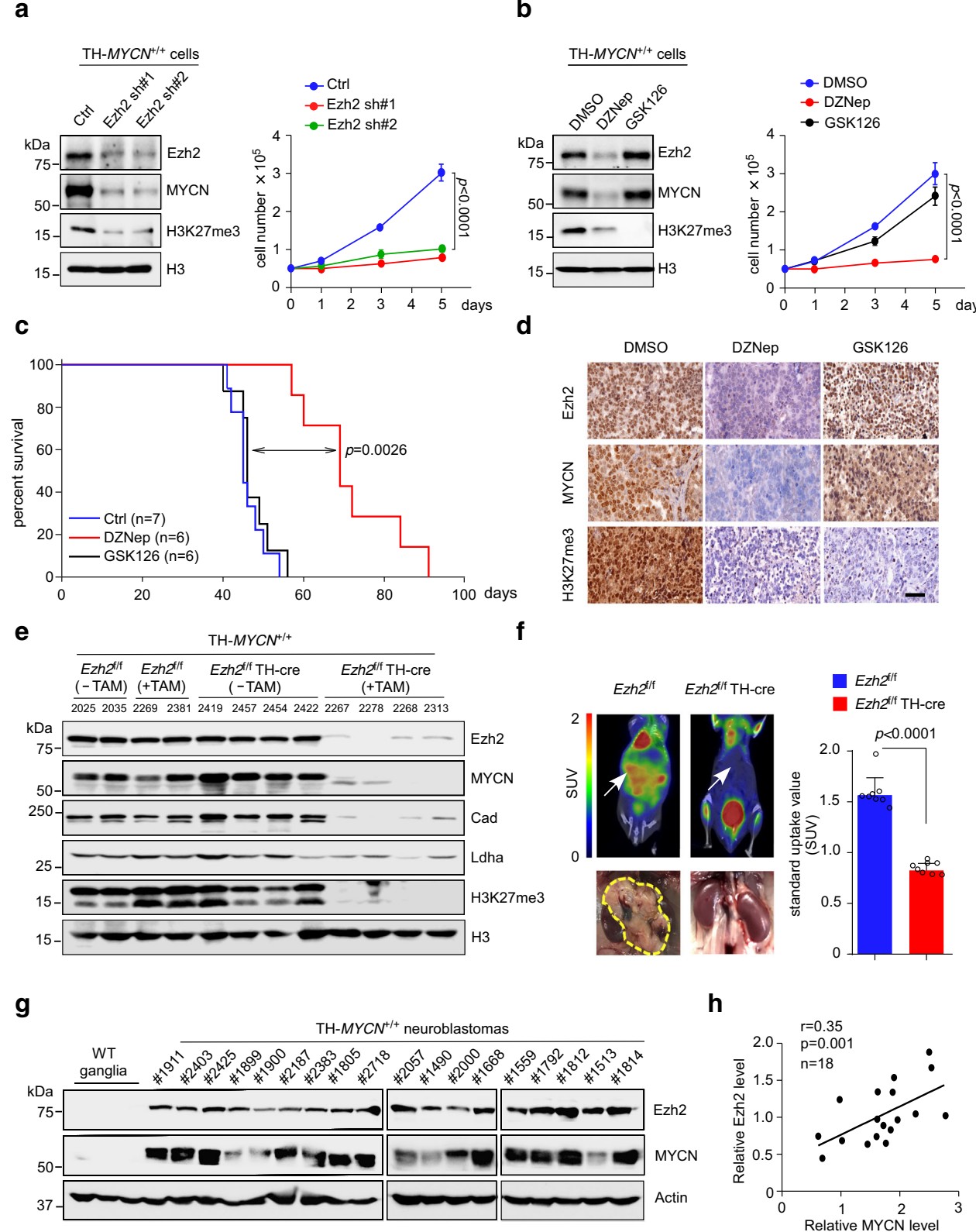

less frequently occurred in human cancers. Further investigating the mechanisms whereby EZH2 promotes cancer is therefore of great significance.

We herein identify a pharmacogenetic interaction between EZH2 as a whole (independent of its enzymatic activity) and oncogenic MYC(N), predicting that the enzymatic inhibitors now in clinical development may be ineffective in MYC(N)-driven

cancers unless they are capable of depleting EZH2 and MYC(N) or disrupting EZH2-MYC(N) interaction. In support of this notion, multiple previous studies observed that tumor cells obtained from Burkitt lymphoma, double-hit B cell lymphoma, and triple-negative breast cancer are largely insensitive to EZH2 enzymatic inhibitors[9,37,38]. Of particular note, MYC is also frequently deregulated in these cancer types and meanwhile these

**Fig. 6 Ezh2 conditional knockout effectively depletes MYCN and impedes tumor formation of MYCN-driven neuroblastoma. a, b** Analysis of MYCN expression and cell proliferation upon shRNA depletion of murine Ezh2 (**a**) or administration of 5 μM of DZNep or GSK126 (**b**) in primary neuroblastoma cells from TH-MYCN$^{+/+}$ mice. Data shown were obtained from averages (±SD) of technical triplicates; significance was determined by two-way ANOVA test followed by Tukey's correction (**a, b**). **c** Kaplan–Meier survival plot of tumor-bearing TH-MYCN$^{+/+}$ mice treated with DZNep (3 mg/kg) or GSK126 (50 mg/kg). Significant difference between the DZNep-treated group versus control was determined by log-rank test. **d** Immunohistochemistry detection of Ezh2, MYCN, and H3K27me3 upon DZNep or GSK126 treatment. Representative stains of indicated proteins are shown using paraffin-embedded tumor sections derived from (**c**). The scale bar represents 50 μm. **e** Immunoblots of the indicated proteins in primary sympathetic ganglia lysates from Ezh2-deleted TH-MYCN$^{+/+}$ mice and age-matched controls. Histone H3 was used as a loading control. **f** Representative FDG-PET images of murine neuroblastoma (left) and quantification of the standard glucose uptake in Ezh2-deleted TH-MYCN$^{+/+}$ mice and age-matched controls ($n = 8$ per group). Graph shows mean ± SD; significance was determined by unpaired two-tailed Student's t-test. **g** Immunoblots of Ezh2 and MYCN in TH-MYCN$^{+/+}$ primary neuroblastoma tumor lysates. **h** Correlation between Ezh2 and MYCN protein levels in mouse neuroblastoma samples as shown in (**g**). The experiments were independently repeated three times with similar results (**d–e, g**). Source data are provided as a Source data file.

tumor cells rely on high levels of MYC for survival. In addition to MYC, EZH2 was previously shown to promote the expression of NF-κB targets and tumor cell growth independent of its histone methyltransferase activity in ER-negative basal-like breast cancer[39].

Overall, our results highlight the need for development of next generation of EZH2 inhibitors specifically targeting the EZH2 protein as a whole (e.g. design EZH2 pharmacologic degraders through PROTAC-based technology) or disrupting EZH2-MYC(N) protein interactions, as patients with MYC(N) deregulation are likely to respond.

## Methods

**Cell culture and reagents**. 293T (MCRL-3216), NCI-H526 (CRL-5811), NCI-H2171 (CRL-5929), NCI-H82 (HTB-175), and Daudi (CCL-213) cells were purchased from American Type Culture Collection (ATCC). Human neuroblastoma cell lines Kelly, SK-N-BE2, SMS-KAN, SK-N-AS, SH-SY5Y, NBLS, SHEP, and BE-2C cells were kindly provided by Drs. John M Maris and Michael D. Hogarty (Children's Hospital of Philadelphia, University of Pennsylvania, Philadelphia, PA, USA). 293T cells were maintained in Dulbecco's modified Eagle's medium (DMEM, Hyclone) containing 10% fetal bovine serum (FBS, Gibco) and 1% penicillin/streptomycin (Hyclone). Kelly, SK-N-BE2, SMS-KAN, SK-N-AS, SH-SY5Y, NBLS, SHEP, BE-2C, NCI-H526, NCI-H82, and Daudi cell lines were grown in complete RPMI-1640 (Hyclone) supplemented with 10% FBS and 1% penicillin/streptomycin. NCI-H2171 cells were cultured in complete RPMI-1640 (Hyclone) supplemented with 20% fetal bovine serum. All cell lines, authenticated with short tandem repeats (STR) analysis, were generally cultured at 37 °C under 5% $CO_2$ for fewer than 6 months after resuscitation and regularly tested for mycoplasma contamination using MycoAlert (Lonza). DMSO (Sigma-Aldrich), Cycloheximide (CHX, Sigma-Aldrich, 508739), 3-deazaneplanocin A hydrochloride (DZNep, MCE, HY-12186), GSK126 (Selleck, 7061), EPZ6438 (Tazemetostat, Selleck, 7128), and MG132 (Selleck, 2619) were used in cell treatments.

**Primary cell culture**. Isolation and preparation of TH-MYCN$^{+/+}$ primary tumor cells were conducted as previously described[40]. Tumor tissues with a volume of 0.5 cm³ were dissected from TH-MYCN$^{+/+}$ transgenic mice, washed with phosphate-buffered saline (PBS) before minced, and digested with 0.25% trypsin (Sigma) for 15 min. The resulting cell mixture was washed twice with PBS and treated with red blood cell (RBC) lysis buffer (Biolegend) to remove debris before centrifuged for collecting primary neuroblastoma cells. As suggested in the previous report[41], primary tumor cells were cultured with DMEM/F12ham (Hyclone) supplemented with 15% FBS (Gibco), 2% B-27 supplement (Gibco), 1% penicillin/streptomycin (Hyclone), 1% non-essential amino acids (Gibco), 1 mM sodium pyruvate (Sigma), 10 ng/mL epidermal growth factor (EGF, Peprotech), 15 ng/mL basic fibroblast growth factor (bFGF, Peprotech), and 55 μM β-mercaptoethanol (Sigma).

**Mass spectrometry analysis**. Total cellular proteins extracted from Kelly and NCI-H2171 cells were subjected to immunoprecipitation (IP) using MYCN and MYC antibodies. Immunoprecipitated proteins were washed, and eluted with 0.2 M glycine pH 2.0. Eluted proteins were precipitated with acetone and digested with trypsin. The resulting peptides were desalted on the C18 Stage tips (Thermo Fisher Scientific) and loaded onto an EASY-nLC system and analyzed in an Orbitrap Exploris 480 mass spectrometer equipped with the FAIMS Pro interface (Thermo Fisher Scientific). Full mass spectrometry resolutions were set to 60,000 at m/z 200 and mass range was set to 350–1500. Raw files were processed with Proteome Discoverer 2.4 (Thermo Fisher Scientific). Two-sided t-tests were performed to compare binding protein abundance between IgG and IP groups. Candidate proteins were defined by the following criteria: p value < 0.05 with a fold

change >2. The complete lists of quantified proteins are shown in Supplementary Data 1 and Supplementary Data 2.

**Immunoblot and immunoprecipitation**. For immunoblot, cells were lysed in RIPA buffer (50 mM Tris-HCl pH 7.4, 150 mM NaCl, 1% Triton X-100, 1% sodium deoxycholate, 0.1% SDS, 2 mM sodium pyrophosphate, 25 mM β-glycerophosphate, 1 mM EDTA, 1 mM $Na_3VO_4$, and 0.5 μg/mL leupeptin) and protein concentrations were determined using BCA assay kit (Thermo Fisher Scientific). In all, 30–50 μg total cellular proteins were subjected to SDS-PAGE and transferred to polyvinylidene difluoride (PVDF) membrane (Bio-Rad). Blots were generally blocked with 5% fat-free milk for 1 h before incubated with primary antibodies at 4 °C overnight. Appropriate horseradish peroxidase (HRP)-conjugated secondary antibodies were applied for 1 h at room temperature. Protein bands were detected with SuperSignal Chemiluminescent Substrate (Bio-Rad) and visualized using Chemi Doc$^{TM}$ Touch Imaging System (Bio-Rad). Densitometric analysis of protein abundance was determined by ImageJ software. For immunoprecipitation (IP), cells were lysed in IP buffer (50 mM Tri-HCl pH 7.4, 150 mM NaCl, 1 mM EDTA, 1% NP-40, 1 mM DTT, and protease inhibitor cocktail). The resulting cell lysates (1 mg) were incubated with corresponding antibodies conjugated to Protein G beads (Thermo Fisher Scientific) overnight at 4 °C. Immunoprecipitated proteins were washed three times with IP buffer, and then subjected to SDS-PAGE and immunoblot. Primary antibodies are listed in Supplementary Table 1.

**In vitro protein interaction assay**. His-tagged MYCN protein was purchased from CUSABIO (CSB-YP015278HU). DNA sequences encoding EZH2, SUZ12, and EED were cloned into the pGEX-4T vector, in frame with glutathione S-transferase (GST). Recombinant proteins were expressed in BL21(DE3) Escherichia coli cells, and GST-tagged proteins were purified and immobilized onto glutathione-agarose beads (PerkinElmer). GST beads were incubated overnight with His-tagged MYCN. Purified complexes were washed three times with IP lysis buffer, then separated on SDS-PAGE and analyzed by Coomassie staining and/or immunoblot analysis.

**Chromatin immunoprecipitation sequencing (ChIP-seq)**. SK-N-BE2 ($1 \times 10^7$) cells were fixed with 1% paraformaldehyde in PBS for 10 min and then quenched with 0.125 M glycine for 5 min at room temperature. Fixed cells were sequentially lysed with 10 mL lysis buffer 1 (50 mM HEPES-KOH, pH 7.5, 140 mM NaCl, 1 mM EDTA, 10% glycerol, 0.5% NP-40, 0.25% Triton X-100 and 1× protease inhibitors) and 10 mL lysis buffer 2 (10 mM Tris-HCl, pH 8.0, 200 mM NaCl, 1 mM EDTA, 0.5 mM EGTA and 1× protease inhibitors). Cell lysates were then resuspended in 300 μL lysis buffer 3 (10 mM Tris-HCl, pH 8.0, 100 mM NaCl, 1 mM EDTA, 0.5 mM EGTA, 0.1% Na-deoxycholate, 0.5% N-lauroylsarcosine and 1× protease inhibitors), and sheared using the Diagenode Bruptor Plus with the high-power mode for 25 cycles (sonication cycle: 30 s ON, 30 s OFF). One percent Triton X-100 was added to the samples which were centrifuged at 20,000g for 10 min to remove the insoluble debris. Sonicated human chromatin was spiked-in with 10–30% mouse chromatin for normalization[42]. DNA was immunoprecipitated with 10 μg individual antibodies and 15 μL pre-blocked Protein A/G beads (Smart-Lifesciences), and eluted with 200 μL elution buffer (50 mM Tris-HCl, pH 8.0, 10 mM EDTA, 1.0% SDS, and 200 μg/mL proteinase K) prior to phenol–chloroform extraction and ethanol precipitation.

Library preparation was done using the NEBNext ultra II DNA library prep kit for Illumina, followed by sequencing on a NovaSeq 6000. ChIP-seq reads were aligned to the human genome (UCSC hg38) with Bowtie2, allowing only uniquely mapping reads with up to two mismatches in the 150 bp reads. The aligned human BAM files were normalized to the total aligned reads and converted to bigwig files for visualization in the UCSC Genome Browser. Peaks were called using MACS2 (model-based analysis of ChIP-seq) version 2.1.2 with default parameters. Sonicated human chromatin was spiked-in with 10–30% mouse chromatin for normalization. DeepTools were used to generate heatmap from bigwig files.

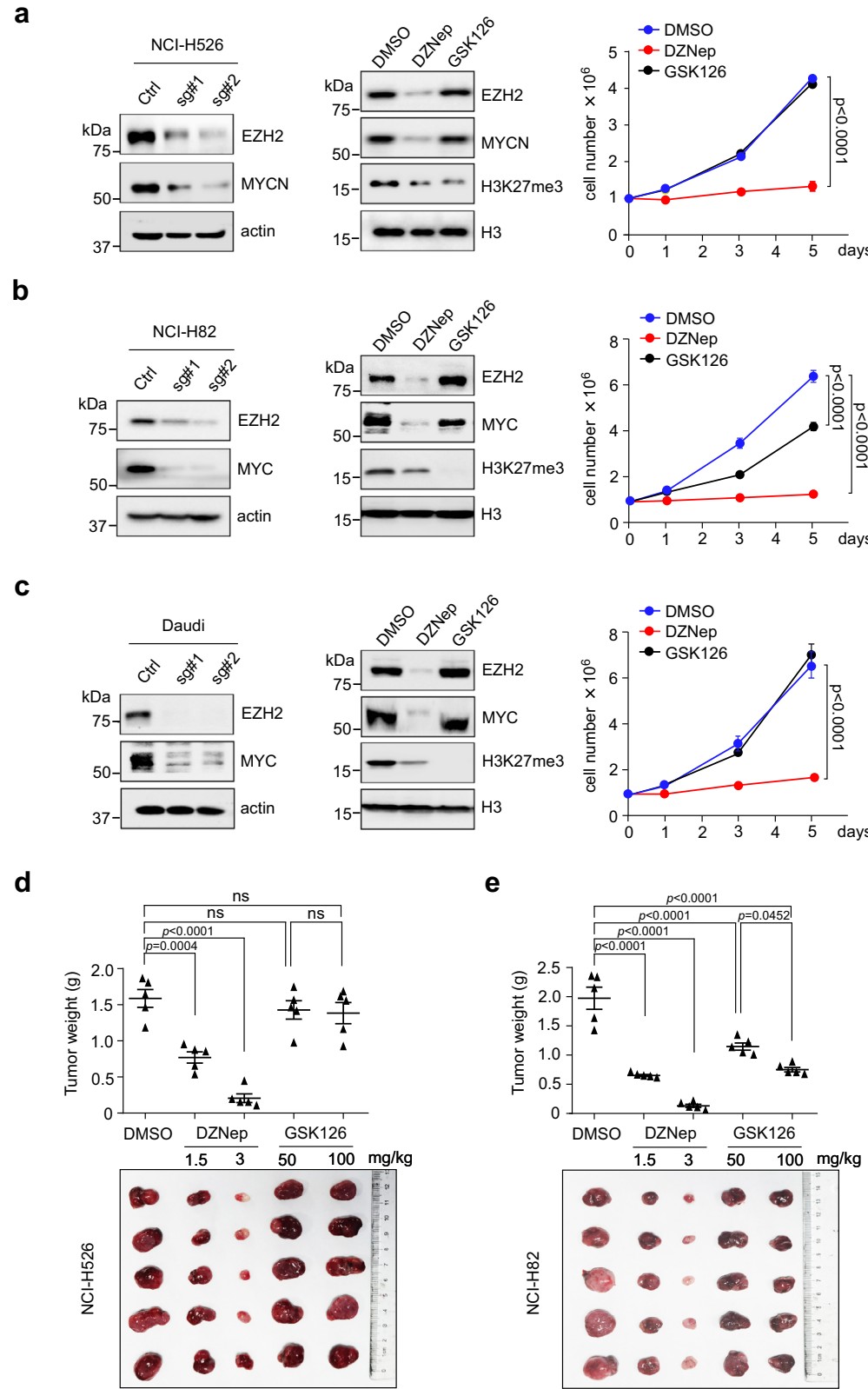

**Chromatin immunoprecipitation.** SK-N-BE2 cells were fixed with 1% paraformaldehyde at room temperature for 10 min, then quenched with 0.125 M glycine for 5 min and lysed in SDS lysis buffer. Cell lysate was subjected to a Bioruptor Pico Sonifier to shear chromatin DNA to a size range of 500–1000 bp. Precleared chromatin was immunoprecipitated with antibody against MYCN or mouse IgG for 16 h at 4 °C. Antibody–chromatin complexes were pulled down with protein G agarose/salmon sperm DNA beads (Roche) (1 h, 4 °C). The eluted DNA was purified and quantified by qPCR using specific primers listed in Supplementary Table 2.

**RNA isolation and real-time qPCR.** Total cellular RNAs were prepared using TRIzol (Thermo Fisher Scientific). RNAs (1 µg) were reverse transcribed with RevertAid first-strand complementary DNA synthesis kit according to the manufacturer's instructions (TOYOBO). Quantitative PCR was performed using FAST SYBR Green Master Mix on CFX Connect Real-Time PCR System (Bio-Rad). Relative mRNA expression was calculated by the $2^{-\Delta\Delta Ct}$ method and normalized to *ACTIN*. Specific PCR primer sequences are listed in Supplementary Table 2.

**Fig. 7 Depletion of EZH2 reduces MYC(N) expression and suppresses growth of MYC(N)-driven small cell carcinoma cells and MYC-driven Burkitt's lymphoma cells. a–c** Analysis of MYCN or MYC expression and cell proliferation upon EZH2 depletion by specific sgRNAs or administration of 5 μM of DZNep or GSK126 in NCI-H526 (**a**), NCI-H82 (**b**), or Daudi (**c**) cells. Actin or H3 was used as a loading control. Data shown were obtained from averages (± SD) of technical triplicates; significance was determined by two-way ANOVA test followed by Tukey's correction. The experiments were independently repeated three times with similar results (**a–c**). **d, e** Xenograft tumor growth assays were performed using NCI-H526 (**d**) and NCI-H82 (**e**) cells upon DZNep or GSK126 treatment. Images of subcutaneous tumors (*n* = 5 per group) under different treatments are shown. Data shown were obtained from averages (±SD) of technical triplicates; significance was determined by one-way ANOVA test followed by Tukey's correction (**d, e**). Source data are provided as a Source data file.

**Lentiviral transduction**. Lentiviral production and transduction were carried out as described[43]. Briefly, lentiviral vectors (pLKO.1 for shRNA and pHAGE for overexpression) were used for plasmid construction and transfected into 293T cells together with packaging plasmids (pMD2.G and psPAX2) using Lipofectamine 2000 (Thermo Fisher Scientific). Viral supernatants were generally harvested 48 h later. For lentiviral transduction, one million neuroblastoma cells or small-cell lung cancer cells were incubated with 0.5 mL viral supernatant and 8 μg/mL polybrene (Sigma) in a final volume of 2 mL for 20 h. Transduced cells were subjected to puromycin (1.5 μg/mL)-mediated selection for additional 20 h. Cell viabilities or gene expression changes were assessed 40 h post-infection.

**Time-course analysis of MYCN degradation**. Cycloheximide (CHX) pulse-chase experiments were conducted to determine MYCN protein half-life[44]. Cells were treated with CHX (50 μg/mL) and then harvested at specific time-points. Total cell lysates were seperated by SDS-PAGE and protein levels were analyzed by immunoblot. MYCN protein band densities were quantified by ImageJ software.

**Ubiquitination analysis**. Cells were lysed in 100 μL 1% SDS lysis buffer (50 mM Tris-HCl, pH 7.4, 150 mM NaCl, 1% NP-40, 1% SDS, 1 mM EDTA). Cell lysates were denatured at 95 °C for 10 min to disrupt protein interaction and then diluted with 900 μL IP buffer, and subjected to centrifugation at 12,000*g*. A small portion of supernatant was saved as input to detect protein expression, and majority of cell extract underwent immunoprecipitation with specific antibodies. MYCN protein polyubiquitination was examined by immunoblot analysis with anti-ubiquitin antibody.

**In vitro protein methylation assay**. Protein methylation assay was performed as described[45]. One microgram of recombinant histone octamers (BPS Bioscience), together with 0.5 μg of recombinant EZH2/EED/SUZ12/RbAp48/AEBP2 complex (PRC2 WT, BPS Bioscience) or EZH2(H689Y)/EED/SUZ12/RbAp48/AEBP2 complex (PRC2 MUT, BPS Bioscience), was incubated in the methylation assay buffer (50 mM Tris pH 8.5, 20 mM KCl, 10 mM DTT, 250 mM sucrose) for 1 h at 30 °C in the presence of *S*-[methyl-$^3$H] adenosylmethionine (PerkinElmer). Reactions were stopped by addition of SDS loading buffer, and samples were then heated for 5 min at 95 °C before separated by Criterion$^{TM}$ Tris-HCl Precast Gel (4%-15%) and then subjected to autoradiography.

**Mice**. BALB/c nude mice were purchased from Beijing Vital River Laboratory Animal Technology Co., Ltd. TH-*MYCN*$^{+/-}$ mice on the 129X1/SvJ genetic background were provided as a courtesy by Prof. William Weiss (University of California-San Francisco, CA, USA). Mice carrying a conditional knockout of *Ezh2* alleles Ezh2$^{f/f}$ (Ezh2$^{tm2Sho}$/J, #022616) and transgenic mice expressing a tamoxifen-inducible cre recombinase under the control of tyrosine hydroxylase (TH) promoter TH-cre$^{ERT2+/-}$ (Th$^{tm1(cre/Esr1)Nat}$/J, #008532) were purchased from Jackson Laboratory (JAX). Both strains are on a mixed C57BL/6 and 129 background. All mice were maintained in Specific Pathogen Free (SPF) animal facility of Medical Research Institute, Wuhan University. Mice were housed in groups of 4–6 mice in an individually ventilated cage (IVC) in a 12:12 light–dark cycle (08:30–20:30 light; 20:30–8:30 dark). The ambient temperature was 22 ± 2 °C with 50–60% relative humidity. All animal experiments were performed according to animal ethical regulations and with approval from the Institutional Animal Care and Use Committee of Wuhan University.

**xenografts**. Female BALB/c nude mice (6 weeks old) were injected subcutaneously with cancer cells resuspended in 200 μL PBS containing 50% Matrigel (v/v). Once tumors reached an average volume of 100 mm$^3$, mice were randomly divided into nine groups undergoing the following treatment, Ctrl (saline), DZNep (0.75, 1.5, 3, and 6 mg/kg), GSK126 (25, 50, 100, and 150 mg/kg) for consecutive 2 weeks. DZNep was intraperitoneally injected into mice every other day and GSK126 once each day. Tumor volumes were measured every 2 days using calipers, and calculated using the formula: length × width$^2$ × 0.5. Upon treatment termination, mice were euthanized and subcutaneous tumors were resected, followed by tumor weights assessment. The maximum tumor-bearing mouse does not exceed 1000 mm$^3$. All animal experiments were performed according to animal ethical

regulations and with approval from the Institutional Animal Care and Use Committee of Wuhan University.

**Generation and genotyping of transgenic mice**. *Ezh2*$^{f/f}$ or TH-cre$^{ERT2}$ hetero-zygote mice on a 129 and C57BL/6 mixed background were backcrossed two generations to the 129X1/SvJ background. *Ezh2*$^{f/f}$ mice were then crossed with TH-*MYCN*$^{+/-}$ mice to generate double transgenic TH-*MYCN*$^{+/-}$*Ezh2*$^{f/f}$ mice. To generate the TH-*MYCN*$^{+/+}$ *Ezh2*$^{f/f}$ TH-*cre* strain, TH-*cre*$^{ERT2}$ mice were mated with TH-*MYCN*$^{+/-}$*Ezh2*$^{f/f}$ mice, as illustrated in Fig. S6d. Genotyping was performed as previously described[33,34,46] using gene-specific PCR primers listed in Supplementary Table 2.

**Neuroblastoma allografts**. Tumor slices (an average of 5 mm$^3$) from TH-*MYCN*$^{+/+}$ transgenic mice were directly subcutaneously inoculated into syngeneic wide type 129X1/SvJ mice (male 6 weeks old). When tumors reached an average volume 200 mm$^3$, mice were randomly divided into five groups for treatment: Ctrl (saline), DZNep (1.5 mg/kg), GSK126 (50 mg/kg), and GSK126 (100 mg/kg). DZNep was intraperitoneally injected into mice once every 2 days and GSK126 once every day. Tumor volumes were measured every 2 days and tumor weights were assessed in sacrificed animals.

**Positron emission tomography (PET) imaging**. 18-Fluoro-6-deoxy-glucose (FDG) (200 μCi) was intravenously injected into mice that had been fasted for 12 h. One hour later, mice were subjected to PET/CT scans and images were acquired by the TransPET Discoverist 180 system (Raycan Technology Co., Ltd, Suzhou, China). PET images were reconstructed using the three-dimensional (3D) OSEM method with a voxel size of 0.5 × 0.5 × 0.5 mm$^3$. CT images were reconstructed using FDK algorithm with 1024 × 1024 × 1024 matrix. Images were then displayed using the Carimas software (Turku PET Center, Turku, Finland). The mean standardized uptake value (SUV) was calculated by dividing the mean FDG activities by the injected dose and animal weight.

**Immunohistochemistry (IHC)**. Tumor tissue sections were deparaffinized, and then rehydrated through an alcohol series followed by antigen retrieval with sodium citrate buffer. These sections were blocked with goat serum in PBS for 1 h at room temperature before incubation with specific antibodies at 4 °C overnight. Slides were then subjected to HRP-linked secondary antibodies for 1 h at room temperature and staining was visualized by the DAB substrate kit (Vector Labs). Representative IHC images were captured at ×400 magnification and quantified by ImageJ software.

**Quantification and statistical analysis**. Statistical analysis were carried out using GraphPad Prism 7. Comparisons of two groups were analyzed using unpaired two-tailed Student's *t*-test, and statistical significance from three or more groups were calculated by one-way or two-way ANOVA with Tukey's corrections. Survival of tumor mouse model was presented as Kaplan–Meier curves and significance was estimated by log-rank test. Differences were considered significant when *p* < 0.05.

**Reporting summary**. Further information on research design is available in the Nature Research Reporting Summary linked to this article.

## Data availability

The mass spectrometry proteomics data generated in this study have been deposited in the ProteomeXchange database under accession code PXD029652. The ChIP-seq data generated in this study have been deposited in the NCBI database under accession code GSE181782. The remaining data are available within the Article, Supplementary Information or Source Data file. Source data are provided with this paper.

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

## Acknowledgements

We thank members of Qing and Liu Laboratories for helpful suggestions. We also thank Dr. John M. Maris (Children's Hospital of Philadelphia, University of Pennsylvania, Philadelphia, PA, USA) for generously sharing human neuroblastoma cell lines, Dr. William Weiss (University of California-San Francisco, CA, USA) for kindly providing TH-*MYCN* mice. This study was supported by grants from National Key R&D Program of China (2017YFA0505600 to G.Q.), National Natural Science Foundation of China (81830084 to G.Q.), National Science Foundation for Distinguished Young Scholar (81725013 to G.Q. and 82025003 to H.L.), Hubei Provincial Natural Science Fund for Creative Research Groups (2021CFA003 to H.L.), and Seed Fund Program for Sino-Foreign Joint Scientific Research Platform of Wuhan University (to G.Q.).

## Author contributions

G.Q. conceived and designed the study. G.Q. and H.L. supervised the study. G.Q., L.W., and H.L. wrote the manuscript. L.W. performed most of the experiments. C.C., Z.S., and W.K. provided technical assistance in protein interaction assay, performed mass spectrum analysis and mouse model study. M.Y. conducted molecular cloning and made various plasmid constructs. H.W. and D.W. performed ChIP-seq analysis. All authors contributed to the final manuscript.

## Competing interests

The authors declare no competing interests.
