## [Peer Review File · Nature Communications]

EZH2 depletion potentiates MYC degradation inhibiting neuroblastoma and small cell carcinoma tumor formationReviewers' Comments:

Reviewer #1:

Remarks to the Author:

Wang et al. show in their manuscript "EZH2 depletion potentiates MYC degradation and immunotherapy efficacy" that EZH2 can stabilize MYC proteins. Degradation, but not inhibition of EZH2's methyltransferase activity, results in polyubiquitination and proteasomal degradation of MYC. Decreased EZH2 levels and the resulting decrease in MYC inhibit growth of tumor cells in MYCN-driven neuroblastoma (NB) as well as in MYC-driven small-cell lung cancer. Mouse models of NB show a significant survival benefit upon treatment with DZNep, which depletes EZH2 protein as a whole. Additionally, interfering with this axis results in increased PD-L1 level. Due to this increase tumor are sensitized to the immune checkpoint inhibitor α -PD1. In my opinion the study is potentially interesting, but there are three major concerns:

- First, there is virtually no information about statistics and almost all panels lack information about repeats. For example, all proteomic data need to be shown as Volcano plots with p-values derived from repeats on the y-axis and ratio to IgG as the x-axis and all stability data need error bars that show SD, for example. In general, all data need better information about statistics.
- Second, the interaction of MYCN with EzH2 has been previously characterized in great detail (1). This paper has shown that MYCN drives an EzH2-dependent repressive (see below) transcriptional program in MYCN-dependent neuroendocrine tumors. This paper is not quoted by the authors and it should have been. The dependence of MYCN-amplified tumors on EzH2 has also been well documented in multiple papers (2). Both observations compromise the novelty of this manuscript. The critical novelty that would carry the paper, in my view, would be the demonstration that this is direct: and this is, despite the claim, not shown. So in my view the authors need to demonstrate that a component of the PRC2 complex recognizes the phosphor-degron directly (or stabilizes Aurora-A on the degron), clarify which component it is and what exactly it recognizes and show that it competes with FBXW7: these experiments are technically feasible.
- Third, PRC2 is generally a repressive complex so it is not at all clear why this should stabilize MYCN, which is mainly activates transcription. In the Dardenne study mentioned above, EzH2 is required to suppress a specific program of cellular differentiation. So the authors need either perform a ChIPseq to show whether MYCN is removed from specific promoters if PRC2 is gone or an RNAseq to show that genes that are repressed by MYCN are activated.

Specific Comments

1. It is not clear how the immunoprecipitations were carried out and how the authors ruled out that the interactions were mediated indirectly by DNA or RNA. Appropriate controls are needed.
2. By now, multiple interactomes of MYC and MYCN have been published and they show a very high degree of overlap (3). The fact that – in the authors hands - there is such a limited overlap suggests that there is a high amount of non-specific background in the IPs. In any case, they should show a Venn diagram how their interactomes relate to the interactomes from the Penn, Eilers and Beli laboratories.
3. The idea that EzH2 competes with Fbxw7 is interesting. This mechanism has been published before for Aurora-A (4,5). The critical piece of evidence missing here is – despite the claim in Figure 1- that EzH2 binds the phosphodegron in MYCN directly and competes off FBXW7 in an in vitro reaction. In vitro experiments will also allow the authors to test directly the specificity of degron recognition by EzH2. The authors should also use both in vitro and cellular assays to test whether – consistent with Dardenne et al.- EzH2 and Aurora-A form a joint complex and it is the interaction with Aurora-A that stabilizes MYCN.
4. The authors claim that the described mechanism is conserved between MYC and MYCN. I wondered, especially for NB, if EZH2 degradation is specific for MYCN-amplified NB or if it is also effective in MYC-driven NB. To clarify the authors could test a panel of MYCN-amplified vs. MYCN-non-amplified NB cell

lines, since most of the MYCN-non-amplified cells show high MYC levels.

5. It is unclear why the authors chose Kelly cells as their major NB cell line. Kelly cells are p53 mutant which doesn't reflect the majority of NB cell lines. During their study they add some other NB cell lines but I think some of the early results should be repeated with a p53 wildtype MYCN-driven NB cell line as well.

6. In line 90 the authors claim that sgRNA and shRNA give similar results. In my opinion this statement does not reflect what the authors show. Since with sgRNA only protein levels are affected while shRNA decreases both mRNA as well as protein levels.

7. Throughout the paper the authors switch between different nomenclature of MYC making it sometimes difficult to understand if the conclusion is for c-MYC or for all MYC proteins.

8. In line 117 the authors talk about inhibition of FBW7 – maybe it would be better to use “shRNA-mediated knockdown” instead.

9. In line 160 the authors clarify that catalytic inhibitors don't affect the protein levels of other PRC2 components. What about DZNep? Is it specific in decreasing EZH2 levels or does it also decrease levels of other PRC2 components?

10. Within the manuscript the authors switch between FBW7 and FBW7 α - What about the other isoforms? If the mechanism is true only for the one isoform the authors might consider using only FBW7 α throughout the manuscript.

11. The manuscript clearly shows the differences between DZNep and the catalytic inhibitor GSK126. In Figure 4 e-h the animals were treated with both inhibitors. 4e shows that tumor weight is impaired and that the doses to use are very different. They claim that the effect of 150 mg/kg GSK126 is off-target which is plausible. But still it is not how doses were chosen for the survival study? Why use 50 mg/kg and not 100 mg/kg?

12. In their last Figure the authors show that interfering with the EZH2-MYC axis increases PD-L1 and that therapy can be improved by adding α -PD1. Since NB are immunological cold tumors and the mouse model used has also only little immune cell infiltration it would be nice to see if immune cell infiltration is changed after treatment with DZNep and also with the combination of DZNep and α -PD1.

13. All cell culture experiments were done in a MYCN-driven neuroblastoma cell line as well as in a MYC-driven small-cell lung cancer cell line. Nevertheless the in vivo part in mice was only performed in different NB models but not in any lung cancer models. In the end, the authors conclude that the therapeutic strategy could be applied for MYC-driven tumors. To make this point stronger I think the authors should show that the same or a similar effect is also true for the c-MYC driven lung cancer. Otherwise maybe they have to be more speculative if the mechanism is really conserved.

14. In my opinion some graphs are not so clear and would benefit from using colors instead of different symbols and line styles. This is for example the case in Figure 4 a and b. Since the Figure is in color already the authors might consider changing to make it easier for the reader to distinguish different conditions.

References

1. Dardenne E, Beltran H, Benelli M, Gayvert K, Berger A, Puca L, et al. N-Myc Induces an EZH2-Mediated Transcriptional Program Driving Neuroendocrine Prostate Cancer. *Cancer cell* 2016;30(4):563-77 doi 10.1016/j.ccell.2016.09.005.
2. Chen L, Alexe G, Dharia NV, Ross L, Iniguez AB, Conway AS, et al. CRISPR-Cas9 screen reveals a MYCN-amplified neuroblastoma dependency on EZH2. *J Clin Invest* 2018;128(1):446-62 doi 10.1172/JCI90793.
3. Baluapuri A, Wolf E, Eilers M. Target gene-independent functions of MYC oncoproteins. *Nature reviews* 2020;21(5):255-67 doi 10.1038/s41580-020-0215-2.
4. Otto T, Horn S, Brockmann M, Eilers U, Schuttrumpf L, Popov N, et al. Stabilization of N-Myc is a critical function of Aurora A in human neuroblastoma. *Cancer cell* 2009;15(1):67-78 doi 10.1016/j.ccr.2008.12.005.
5. Richards MW, Burgess SG, Poon E, Carstensen A, Eilers M, Chesler L, et al. Structural basis of N-Myc binding by Aurora-A and its destabilization by kinase inhibitors. *Proceedings of the National Academy of Sciences of the United States of America* 2016;113(48):13726-31 doi 10.1073/pnas.1610626113.

Reviewer #2:

Remarks to the Author:

This is a very interesting study focused on analyzing regulators of MYC transcription factor functions through protein-protein interactions. The authors unveil a catalytic-independent function of EZH2 in the stability of MYC-transcription factors. The manuscript is well written and a logical set of experiments is conducted and properly controlled. Multiple methodological approaches (shRNA, CRISP, pharmacological inhibition) support the robustness of their findings. However, some questions remain open and need clarification before the manuscript gets accepted.

Major concerns:

1. The authors claim in the title that EZH2 controls MYC stability in general. This conclusion is withdrawn from experiments carried out in different cancer models where only c-Myc is expressed (Daudi and NCI-H2171 cells) or in models where N-Myc is expressed (Kelly cells). However, most of Neuroblastoma cells co-express N-MYC and C-MYC. Thus, it will be important to distinguish whether EZH2 depletion affects both c-MYC and/or N-Myc in neuroblastoma cells. Thus, the question is, in cells where N-Myc and C-Myc are expressed, EZH2 depletion has the same functional consequences?. And more importantly, the EZH2 depletion effects could be rescued by only overexpressing c-MYC or N-MYC?

2. MYCN-amplification is a genetic abnormality that dictates poor prognosis in Neuroblastoma and also other embryonal tumors. However, some of these tumors are also capable of sustaining high MYCN levels without a genetic alteration. The effect of EZH2 on the stability of MYCN are also happening in non-MYCN amplified cell lines?

3. Conversely, there are some neuroblastoma cell lines that do not express MYCN (e.g. SH-SY5Y). To what extent this lack of MYCN expression can be EZH2-dependent? Please discuss and test whether EZH2 is sufficient to stabilize MYCN in cell lines with low or absent levels of MYCN.

4. -It seems clear that EZH2 depletion results in a reduction on MYCN levels. When EZH2 is depleted, what are the consequences to other key elements of the PRC2 complex? Can the levels of MYCN be restored (after EZH2 depletion) by the ectopic expression of other PRC2 components (e.g. SUZ12)?? Please discuss.

5. -Regarding the correlation between EZH2 and MYCN protein levels in neuroblastoma, is this correlation maintained in all analyzed stages, or in patients with and without MYCN-amplification?? Please, clarify. Is any of these correlations also observed at the mRNA level? Please, discuss.

6. -Regarding the in vivo experiments with anti-PD1, the authors should demonstrate whether EZH2 reduction in vivo, causes an increase in the expression of PD1 (by immunohistochemistry) and how this impact in the infiltration of immune cells in the tumor. This experiments will help to define and characterize the effectors of this observed immune response.

7. -Since most of the functional analyses have been performed on EZH2 and MYCN in Neuroblastoma models, this should be reflected in the title. Although, the results points towards a general mechanism on MYC factors, the authors do not prove it at all levels.

Minor concerns:

-Molecular weight bands should be indicated in the immunoblots.

Reviewer #3:

Remarks to the Author:

This is a well written paper with a plethora of data that adds to the information on the interaction of EZH2 with MYC. The novel message in this paper is that EZH2 interaction with MYC is through non-canonical methods that are independent of its methyltransferase activity. The observation that EZH2 has functions other than those related to methyltransferase activity has been previously reported, though its specific interactions with MYC have been incompletely evaluated thus far.

Major Concerns:

1. The majority of experiments are related to NMYCN and not CMYC. The large amount of presented data makes it confusing at times to follow which experiments are related to NMYC and CMYC. Strongly recommend that the paper focus on NMYC. Almost identical experiments (though fewer) were performed to assess EZH2-CYMC interactions. Suggest these be summarized in 1-2 sentences after each result section and all data presented as Supplementary Files.
2. In this same vein, throughout the paper, the authors should specify if the presented results refer to C-MYC or N-MYC. There are numerous areas in the paper where this is not clearly specified.
3. A major point that has NOT been addressed by the investigators is the difference between MYCN overexpression and amplification in neuroblastoma. Most of the initial elegant experiments have been carried out in a single MYCN-amplified cell line (Kelly). This has significant implications in the translational analysis of the results. Do EZH2-N-MYC interactions depend on amplification, overexpression or "normal" expression of MYCN. At least some of the initial experiments should be repeated in MYCN-non-amplified cell line to address this issue.
4. The choice of human cell lines for the various experiments also seems somewhat arbitrary. The characteristics of various cell lines should be described and the rationale for their choice explained. (It is only on line 276 that SHEP MYCN-ER is characterized in passing).
5. Data from human subjects has been included, but no mention has been made about human subject protection e.g. IRB approval.
6. The rationale for considering EZH2 inhibition in combination with immune checkpoint blockade (Lines 302-312) has been very cursorily developed. A major reason for the lack of success for this immunotherapeutic approach in NB has been the lack of HLA Class expression in addition to the PD1/PDL1 axis. Similarly lines 296-301 refer to PD-L1 mRNA not protein. In general, the interpretation of data to emphasize a role for EZH2 inhibition for enhancing immunotherapy is not convincing. Recommend that this section be removed from this paper especially since the more established role of EZH2 function in immune effector cells that would need utilization in such a strategy has not been explored.
7. Would be more valuable in the Discussion section to focus on the mechanisms of EZH2-MYC interactions uncovered by the investigators rather than to discuss the potential for immunotherapeutic strategies.

Minor

1. Last paragraph of introduction is more suited to discussion section.
2. Consider expanding figure legends to concisely state the results of the experiment rather than methods alone.
3. Line 197: clarify what ectopic expression means in this context.
4. Line 228: the statement about off-target effects of GSK 126 is speculative and should either be deleted or moved to Discussion
5. Line 265-268; 298-299: These data are cursory and no further detail is available on the human samples tested. Recommend removing data on human tissue samples from this paper.

Reviewer #4:

Remarks to the Author:

The study by e uncovered a new mechanism of EZH2 in promoting tumorigenesis in which EZH2 acts through interaction with Myc and maintains its stability. The study is very interesting as it shows that

this activity of EZH2 does not require its methyltransferase activity. The mechanism is well elucidated with a rigor set of experiments showing EZH2 induces MYC stability through antagonizing MYC ubiquitination. As such small molecules of EZH2 inhibitor such as DZNep, which causes EZH2 degradation as a whole, can deplete MYC protein expression and inhibits MYC-driven cancer cell proliferation. In contrast, the catalytic inhibitors of EZH2 are unable to do this. This finding has important therapeutic implications, as currently, there are no targeted therapies against Myc. The study further provides novel mechanistic insights into the role of EZH2 in tumorigenesis and highlights the importance of developing EZH2 degraders for cancer therapy. The manuscript is well written, and the experiments are well executed with proper controls. Impressively, both in vitro and in vivo models and data from clinical samples have been provided to support the claim.

Major Comments

The last part shows a role of EZH2/MYC in suppressing PDL1. As such, depletion of EZH2 by DZNep is shown to restore PDL1 expression and thus sensitize anti PDL1 treatment. Although this finding is interesting and potentially expands the role of EZH2/MYC in immune regulation, the mechanism for PDL1 expression might be more complex that may involve both canonical histone and non-histone activities. Several questions remain, eg does EZH2 suppress PDL1 expression through histone methylation at PDL1 loci ? Have the enzymatic inhibitors of EZH2 GSK126 been tested in this context? Does H3K27me3 at PDL1 locus show any changes before and after DZNep treatment?

I feel that the data obtained from Figures 1-5 have adequately delivered sufficient novelty and robustness for publication. This review suggests the authors consider taking out the last part of PDL1, which can be further investigated in a separate study.

Minor comments

Some key references about DZNep and methyltransferase-independent role of EZH2 should be included Tan et al., *Genes Develop* 2007, which reported DZNep as the first-in-class EZH2 inhibitor that induce EZH2 degradation. /doi/10.1101/gad.1524107.

Lee et al, *Molecule Cell*, 2011, which is among the first to report non-catalytic activity of EZH2 in cancer

<https://doi.org/10.1016/j.molcel.2011.08.011>

The following is a point-by-point response to the comments. **Please note that changes made to the paper are indicated in red.**

Reviewer #1, with expertise in ubiquitination and cancer involving MYC:

Wang et al. show in their manuscript “EZH2 depletion potentiates MYC degradation and immunotherapy efficacy” that EZH2 can stabilize MYC proteins. Degradation, but not inhibition of EZH2’s methyltransferase activity, results in polyubiquitination and proteasomal degradation of MYC. Decreased EZH2 levels and the resulting decrease in MYC inhibit growth of tumor cells in MYCN-driven neuroblastoma (NB) as well as in MYC-driven small-cell lung cancer. Mouse models of NB show a significant survival benefit upon treatment with DZNep, which depletes EZH2 protein as a whole. Additionally, interfering with this axis results in increased PD-L1 level. Due to this increase tumor are sensitized to the immune checkpoint inhibitor a-PD1. In my opinion the study is potentially interesting, but there are three major concerns:

- First, there is virtually no information about statistics and almost all panels lack information about repeats. For example, all proteomic data need to be shown as Volcano plots with p-values derived from repeats on the y-axis and ratio to IgG as the x-axis and all stability data need error bars that show SD, for example. In general, all data need better information about statistics.

Based on the Reviewer’s comments, we have provided further information on statistics and repeats in revised figure panels and figure legends. Meanwhile, we have repeated the proteomic data based on the Reviewer’s suggestion, and now all proteomic data are shown as Volcano plots with p-values derived from repeats on the y-axis and ratio to IgG as the x-axis (revised Fig. 1). We have also included error bars with standard deviations for all protein stability assays (revised Fig. 3 and supplementary Fig. 3, 4 and 7). Taken together, we have extensively revised the whole manuscript based on the Reviewer’s comments.

- Second, the interaction of MYCN with EzH2 has been previously characterized in great detail (1). This paper has shown that MYCN drives an EzH2-dependent repressive (see below) transcriptional program in MYCN-dependent neuroendocrine tumors. This paper is not quoted by the authors and it should have been. The dependence of MYCN-amplified tumors on EzH2 has also been well documented in multiple papers (2). Both observations compromise the novelty of this manuscript.

We agree with the Reviewer that a previous study from Dardenne and colleagues (Cancer Cell, 2016) showed that MYCN formed a complex with EZH2 to drive the transcriptional repression of downstream targets in neuroendocrine prostate cancer, where EZH2 was shown to execute a tumor-promoting function in a

methyltransferase-dependent manner at the transcriptional level. Please kindly note that in the current study we demonstrate that EZH2 directly interacts with both MYC family oncoproteins, MYC and MYCN, via the MYC homology box 2 domain (MB2) and promotes their stabilization (at the posttranslational level) largely independent of its methyltransferase activity (revised Fig. 1, 3, 4 and 7). Thus, our results identify that a critical function of EZH2 in human cancers, at least in neuroblastoma and small cell carcinoma, is to stabilize MYC-family oncoproteins and sustain their oncogenic functions. Most likely, both mechanisms would act in concert to promote tumor initiation and progression, especially in MYC(N)-driven cancers.

When the current study is undergoing, Chen and colleagues revealed a dependency on EZH2 in *MYCN*-amplified neuroblastomas (J Clin Invest., 2017), yet the mechanisms accounting for this dependency remain largely unknown. Our results demonstrate that stabilization of MYCN by EZH2 is a critical mechanism responsible for EZH2 dependency in *MYCN*-amplified neuroblastoma. Moreover, we also showed that EZH2 promotes MYC(N) stabilization in *MYCN*-nonamplified neuroblastoma cells, as well as in small cell carcinoma cells and Burkitt's lymphoma cells characterized by MYC(N) overexpression due to chromosome amplification or translocation (revised Fig. 2 and 7). Thus, these findings decipher a previously unsuspected mechanism involved in MYC(N) deregulation, and may establish MYC family super-transcription factors as universal mediators in sustaining EZH2 oncogenic programs given that deregulation of MYC(N) occurs in >50% of human cancers.

We are sorry for the inadvertent mistake not citing a previous study from Dardenne and colleagues (Cancer Cell, 2016). We have now quoted this paper in the revised manuscript (please see page 5 in revised text).

The critical novelty that would carry the paper, in my view, would be the demonstration that this is direct: and this is, despite the claim, not shown. So in my view the authors need to demonstrate that a component of the PRC2 complex recognizes the phosphor-degron directly (or stabilizes Aurora-A on the degron), clarify which component it is and what exactly it recognizes and show that it competes with FBXW7: these experiments are technically feasible.

In the current study, we first performed GST-pulldown assays with recombinant His-tagged MYCN and GST-tagged EZH2, EED or SUZ12 proteins, and confirmed that EZH2, but not EED and SUZ12, directly bound MYCN *in vitro* (revised Fig. 1f). We then verified that MB2, but not MB1 and MB3, within MYCN was essential for MYCN-EZH2 interaction (revised Fig. 1h, 1i). Since the MYCN phosphor-degron is located within MB1, but not MB2, EZH2 interacted with MYCN largely independent of the phosphor-degron. In fact, MYCN T50A phosphorylation-inactive mutant (abolished GSK3 β -mediated phosphorylation of MYCN) bound EZH2 as capable as the wild-type MYCN while it failed to bind the ubiquitin E3 ligase FBW7 α where direct MYCN-FBW7 α interaction requires T50 phosphorylation within the degron (revised Supplementary Fig. 3e). These results

suggest that stabilization of MYCN by EZH2 occurs via mechanisms largely independent of T50 phosphorylation. Indeed, deletion of the whole MB1 within MYCN had an undetectable effect on MYCN-EZH2 interaction (revised Fig. 1h, compare lane 3 vs lane 2).

Previous data from Eliers and colleagues showed that Aurora kinase A (AURKA) interacted with both MYCN and FBW7 α , and counteracted FBW7 α -mediated MYCN degradation in neuroblastoma cells, raising the possibility that EZH2 forms a joint complex with AURKA and stabilizes MYCN via AURKA. However, immunoprecipitation assays revealed that EZH2 was absent in the AURKA immunoprecipitates when both proteins were co-expressed in 293T cells, while considerable amounts of EZH2 were present in the MYCN immunoprecipitates at the same conditions (revised Supplementary Fig. 4g, compare lane 3 vs lane 2). Moreover, endogenous AURKA and EZH2 failed to form a detectable complex *in vivo* even though both proteins respectively bound MYCN (revised Supplementary Fig. 4h). As such, ectopic expression of AURKA failed to enhance the interaction between EZH2 and MYCN (revised Supplementary Fig. 4i, compare lane 3 vs lane 2). In addition, EZH2 similarly promoted MYC stabilization even though MYC cannot form a detectable complex with AURKA (revised Supplementary Fig. 7b, 7d and data not shown). All these data argue that EZH2 counteracted FBW7 α -mediated MYCN degradation largely independent of AURKA.

We then investigated how EZH2 regulated MYCN stabilization. We first confirmed that regulation of MYCN stabilization by EZH2 occurs via mechanisms largely independent of direct MYCN methylation (revised Fig. 4 and Supplementary Fig. 4). We therefore examined whether EZH2 displaced FBW7 α from binding to MYCN given EZH2 directly interacted with MYCN but not FBW7 α (revised Fig. 1 and data not shown). As expected, the EZH2 methyltransferase-inactive mutant (EZH2 H689A) was as capable as the WT EZH2 in dose-dependently displacing MYCN from a complex with FBW7 α when all three proteins were co-expressed in 293T cells (revised Fig. 4e, compare lanes 3-5 vs lane 2, and lanes 6-8 vs lane 2). When reconstituted with recombinant proteins *in vitro*, EZH2 also similarly displaced FBW7 α from binding to MYCN in a dose-dependent manner (revised Fig. 4f, compare lanes 4-6 vs lane 3). Most likely, direct binding of EZH2 to MB2 would cause spacial hindrance and interfere with recognition of MB1 by FBW7 α given MB1 and MB2 are closely located within the N-terminus of MYCN (Fig. 1g).

- Third, PRC2 is generally a repressive complex so it is not at all clear why this should stabilize MYCN, which is mainly activates transcription. In the Dardenne study mentioned above, EzH2 is required to suppress a specific program of cellular differentiation. So the authors need either perform a ChIPseq to show whether MYCN is removed from specific promoters if PRC2 is gone or an RNAseq to show that genes that are repressed by MYCN are activated.

As the major H3K27me3 methyltransferase, EZH2 is commonly believed to execute tumor-promoting functions via transcriptional repression of tumor suppressor genes,

although it remains largely undefined whether modulation of H3K27 trimethylation is the prominent mechanism whereby EZH2 promotes cancer. In the current study, we identify a critical function of EZH2 in human cancers is to stabilize MYC-family oncoproteins and sustain MYC(N) oncogenic functions independent of its methyltransferase activity. The implication of EZH2 in MYCN (and MYC) stabilization prompted us to investigate whether its inhibition affected MYCN transcriptional activities. We thus performed ChIP-seq assays in *MYCN*-amplified SK-N-BE2 cells with antibodies recognizing MYCN and RNA polymerase II. ChIP-seq analysis demonstrated that MYCN loss upon EZH2 depletion markedly decreased MYCN genome-wide occupancy while RNA polymerase II occupancy was minimally affected (revised Fig. 2). As such, knockdown of EZH2 significantly decreased the transcription of representative MYCN targets (revised Fig. 2e), arguing that EZH2 plays an essential role in sustaining MYCN transcriptional activities. In support of this notion, conditional deletion of *Ezh2* in mice depleted MYCN and significantly inhibited the expression of representative MYCN targets (revised Fig. 6e). These results support that *Ezh2* is indispensable for sustaining MYCN deregulation and MYCN-driven neuroblastoma *in vivo*.

Specific Comments

1. It is not clear how the immunoprecipitations were carried out and how the authors ruled out that the interactions were mediated indirectly by DNA or RNA. Appropriate controls are needed.

We have repeated the experiments based on the reviewer's suggestion. As shown in revised Fig. 1e and Supplementary Fig. 1c and 1d, co-immunoprecipitation using Kelly and NCI-H2171 cell lysates showed that endogenous EZH2, EED and SUZ12 were all respectively present in MYCN and MYC immunoprecipitates independent of DNA and/or RNA within the cell lysates. In support of this observation, recombinant GST-tagged EZH2 directly bound His-tagged MYCN *in vitro* where nucleic acids were absent (revised Fig. 1f).

2. By now, multiple interactomes of MYC and MYCN have been published and they show a very high degree of overlap (3). The fact that – in the authors hands - there is such a limited overlap suggests that there is a high amount of non-specific background in the IPs. In any case, they should show a Venn diagram how their interactomes relate to the interactomes from the Penn, Eilers and Beli laboratories.

We have repeated the proteomic data based on the Reviewer's suggestion, and now all proteomic data are shown as Volcano plots with p-values derived from repeats on the y-axis and ratio to IgG as the x-axis (revised Fig. 1). A total of 1561 potential interactors were identified in MYC-bound immunoprecipitates, and 1469 in MYCN-bound ones, among which 1059 proteins (~70% of overlap) were common MYC and MYCN interactors (revised Fig. 1a-c). Of note, 54% (796/1469) of the MYCN interactomes was identical to those from Eilers lab (revised Supplementary Fig. 1a), and 41% (645/1561)

and 31% (497/1561) of the MYC interactomes was identical to those from Penn and Beli labs, respectively (revised Supplementary Fig. 1b). In addition, MYC(N) interactomes from Eilers, Penn and Beli labs were only partially overlapped (data not shown). All these data support that the full spectra of MYC(N) interactomes vary depending on both the approaches and the cell lines used for interactome isolation. Conceivably, MYC(N) would associate with distinct proteins in different biological contexts.

3. The idea that Ezh2 competes with Fbxw7 is interesting. This mechanism has been published before for Aurora-A (4,5). The critical piece of evidence missing here is – despite the claim in Figure 1- that Ezh2 binds the phosphodegron in MYCN directly and competes off FBXW7 in an in vitro reaction. In vitro experiments will also allow the authors to test directly the specificity of degron recognition by Ezh2. The authors should also use both in vitro and cellular assays to test whether – consistent with Dardenne et al.- Ezh2 and Aurora-A form a joint complex and it is the interaction with Aurora-A that stabilizes MYCN.

As discussed above, we showed that EZH2 bound MB2, but not MB1 and MB3, within MYCN. Since the MYCN phosphor-degron is located within MB1, but not MB2, EZH2 interacted with MYCN largely independent of the phosphor-degron. In support of this notion, MYCN T50A phosphorylation-inactive mutant bound EZH2 as capable as the wild-type MYCN, and deletion of the whole MB1 within MYCN had an undetectable effect on MYCN-EZH2 interaction. Our data suggest that direct binding of EZH2 to MB2 may cause spatial hindrance and interfere with recognition of MB1 by FBW7 α given MB1 and MB2 are closely located within the N-terminus of MYCN.

Meanwhile, we failed to identify a detectable interaction between EZH2 and AURKA, and ectopic expression of AURKA failed to enhance EZH2-MYCN interaction. In addition, EZH2 similarly promoted MYC stabilization even though MYC cannot form a detectable complex with AURKA (revised Supplementary Fig. 7b, 7d and data not shown). Our data suggest that EZH2 counteracted FBW7 α -mediated MYCN degradation largely independent of AURKA.

Please kindly see pages 2-3 for more details.

4. The authors claim that the described mechanism is conserved between MYC and MYCN. I wondered, especially for NB, if EZH2 degradation is specific for MYCN-amplified NB or if it is also effective in MYC-driven NB. To clarify the authors could test a panel of MYCN-amplified vs. MYCN-non-amplified NB cell lines, since most of the MYCN-non-amplified cells show high MYC levels.

We performed EZH2 depletion in multiple *MYCN*-amplified and -nonamplified neuroblastoma cell lines. Intriguingly, depletion of EZH2 expression by specific sgRNAs led to a marked decline in MYCN protein abundance in both *MYCN*-amplified Kelly, SK-N-BE2, and SMS-KAN cells and *MYCN*-nonamplified NBL5 cells which express high levels of MYCN (revised Fig. 2a). Since most *MYCN*-nonamplified

neuroblastoma cells exhibited high MYC levels, we also depleted EZH2 expression in SK-N-AS, SH-SY5Y and SK-N-SH cells. As expected, sgRNA-mediated EZH2 abrogation caused a significant depletion of MYC expression in all these cell lines (revised Fig. 2a and Supplementary Fig. 2a). Moreover, in SK-N-SH cells which express both high levels of MYC and low levels of MYCN, EZH2 depletion decreased both MYC and MYCN expression (revised Supplementary Fig. 2a). EZH2 depletion consistently abrogated MYC(N) expression in both p53 wild-type and mutant neuroblastoma cells (revised Fig. 2a), arguing that p53 is not involved in this event.

These findings prompted us to examine whether similar regulation occurs in other MYC(N) driven tumor cells. To evaluate this, we performed EZH2 depletion in representative MYC(N)-driven small cell carcinoma NCI-H526 and NCI-H82 cells as well as MYC-driven Burkitt's lymphoma Daudi cells, which are characterized by MYC(N) overexpression due to chromosome amplification or translocation. As expected, depletion of EZH2 expression by specific sgRNAs significantly reduced MYC(N) protein abundance in all these cells (revised Fig. 7a-c). In addition, we also depleted EZH2 in non-small cell lung carcinoma A549 cells and colorectal carcinoma HCT116 cells, which exhibited deregulated MYC expression due to abnormal activation of the upstream signal pathways, and again EZH2 depletion caused a significant decrease in MYC abundance (data not shown).

Taken together, all these data suggest that EZH2 depletion reduces MYC(N) expression not only in neuroblastoma cells, but also in other cancer cells overexpressing either the *MYCN* or *MYC* oncogene.

5. It is unclear why the authors chose Kelly cells as their major NB cell line. Kelly cells are p53 mutant which doesn't reflect the majority of NB cell lines. During their study they add some other NB cell lines but I think some of the early results should be repeated with a p53 wildtype MYCN-driven NB cell line as well.

We chose the Kelly cells as a start for the immunoprecipitation experiments because these cells exhibit very high levels of endogenous MYCN. Accordingly, we performed series of the following experiments in these same cells. We have now repeated our key points in a number of *MYCN*-amplified and -nonamplified neuroblastoma cell lines (revised Fig. 2 and Supplementary Fig. 2). We also extended our findings in other MYC(N) driven tumor cells (revised Fig. 7 and Supplementary Fig. 7). Our results suggest that regulation of MYC(N) stabilization by EZH2 is a global phenomenon in human cancer cells. Moreover, our results argue that this regulation is largely p53 independent. Please kindly see the revised text and revised figures for more details.

6. In line 90 the authors claim that sgRNA and shRNA give similar results. In my opinion this statement does not reflect what the authors show. Since with sgRNA only protein levels are affected while shRNA decreases both mRNA as well as protein levels.

Please kindly note that depletion of EZH2 expression in multiple *MYCN*-amplified and -nonamplified neuroblastoma cells by either specific sgRNAs or shRNAs significantly reduced MYC(N) protein levels while caused minimal changes in *MYC(N)* mRNA levels (revised Fig. 2a and Supplementary Fig. 2d, 3a). As expected, knockdown of EZH2 by specific shRNAs reduced EZH2 expression at both mRNA as well as protein levels. We have reformatted the supplementary Fig. 2d in an attempt to avoid unnecessary confusion.

7. Throughout the paper the authors switch between different nomenclature of MYC making it sometimes difficult to understand if the conclusion is for c-MYC or for all MYC proteins.

We have edited the nomenclature trying to make it easier to follow. Now in the revised manuscript, MYC and MYCN correspond to the protein products of *MYC* and *MYCN* oncogenes, respectively. We are sorry for this inconvenience.

8. In line 117 the authors talk about inhibition of FBW7 – maybe it would be better to use “shRNA-mediated knockdown” instead.

We have edited the text based on the Reviewer's suggestion (please see line 171 of the revised text).

9. In line 160 the authors clarify that catalytic inhibitors don't affect the protein levels of other PRC2 components. What about DZNep? Is it specific in decreasing EZH2 levels or does it also decrease levels of other PRC2 components?

As shown in Supplementary Fig. 4d, administration of DZNep in Kelly cells also decreased SUZ12 and EED levels, consistent with a previous study (Tan et al, *Genes Dev.* 2007) showing that DZNep treatment depleted EZH2 and reduced EED and SUZ12 expression as well. Most likely, depletion of EZH2 caused SUZ12 and EED degradation.

10. Within the manuscript the authors switch between FBW7 and FBW7a - What about the other isoforms? If the mechanism is true only for the one isoform the authors might considered using only FBW7a throughout the manuscript.

We have edited the text using only FBW7 α throughout the manuscript.

11. The manuscript clearly shows the differences between DZNep and the catalytic inhibitor GSK126. In Figure 4 e-h the animals were treated with both inhibitors. 4e shows that tumor weight is impaired and that the doses to use are very different. They claim that the effect of 150 mg/kg GSK126 is off-target which is plausible. But still it is not how doses were chosen for the survival study? Why use 50 mg/kg and not 100 mg/kg?

Please kindly note that treatment of Kelly xenograft tumors with 50 mg/kg of GSK126 achieved almost the same *in vivo* efficacy as 100 mg/kg of GSK126 in terms of

decreasing H3K27me3 levels (revised Figure 5f, compare lane 6 vs lane 7). Actually, even administration of GSK126 at the dose of 25 mg/kg already exhibited significant efficacy (revised Figure 5f, the H3K27me3 immunoblot in lane 5). We thus chose 50 mg/kg of GSK126 for the following survival study with an aim to maximally avoid any potential off-target effects of the drug. Indeed, 50 mg/kg of GSK126 was already as capable as 3 mg/kg of DZNep in diminishing H3K27me3 expression in TH-MYCN primary neuroblastoma tumors (revised Figure 6d, compare the H3K27me3 immunochemistry images between GSK126 and DZNep treatment). Nevertheless, we also established neuroblastoma allografts in syngeneic 129S/v mice using tumor sections from the TH-MYCN transgenic mice. We respectively used 3 mg/kg of DZNep as well as 50 mg/kg and 100 mg/kg of GSK126 for TH-MYCN allograft treatment. Again, administration of 3 mg/kg of DZNep effectively inhibited the tumor growth, whereas these tumors were found to be similarly resistant to both 50 mg/kg and 100 mg/kg of GSK126 treatment (revised Supplementary Fig. 6b).

12. In their last Figure the authors show that interfering with the EZH2-MYC axis increases PD-L1 and that therapy can be improved by adding a-PD1. Since NB are immunological cold tumors and the mouse model used has also only little immune cell infiltration it would be nice to see if immune cell infiltration is changed after treatment with DZNep and also with the combination of DZNep and a-PD1.

The Reviewer raised a critical scientific question. According to the expert opinion from the other reviewers, we have removed this part from the revised manuscript. We are currently investigating these important issues.

13. All cell culture experiments were done in a MYCN-driven neuroblastoma cell line as well as in a MYC-driven small-cell lung cancer cell line. Nevertheless the *in vivo* part in mice was only performed in different NB models but not in any lung cancer models. In the end, the authors conclude that the therapeutic strategy could be applied for MYC-driven tumors. To make this point stronger I think the authors should show that the same or a similar effect is also true for the c-MYC driven lung cancer. Otherwise maybe they have to be more speculative if the mechanism is really conserved.

We performed EZH2 depletion in representative MYC(N)-driven small cell carcinoma NCI-H526 and NCI-H82 cells as well as MYC-driven Burkitt's lymphoma Daudi cells. As expected, depletion of EZH2 expression by specific sgRNAs significantly reduced MYC(N) protein abundance in all these cells with minimal effect on mRNA levels (revised Fig. 7a-c and Supplementary Fig. 7a). Again, administration of DZNep, which depleted EZH2 and MYC(N) (revised Fig. 7a-c), markedly inhibited proliferation of all the tumor cells examined, while administration of the enzymatic inhibitor GSK126 largely exhibited minimal effect (revised Fig. 7a-c). We then established subcutaneous xenografts using NCI-H526 and NCI-H82 cells. Consistent with the *in vitro* findings, administration of DZNep effectively depleted EZH2 and MYC(N) *in vivo* (revised Supplementary Fig. 7e, f), and caused a significant, dose-dependent inhibition of both

NCI-H526 and NCI-H82 tumors (revised Fig. 7d, e). Taken together, these data suggest that EZH2 depletion induces MYC(N) degradation not only in neuroblastoma cells, but also in other cancer cells overexpressing either the MYCN or MYC oncogene. Please see revised text and figures for more details.

14. In my opinion some graphs are not so clear and would benefit from using colors instead of different symbols and line styles. This is for example the case in Figure 4 a and b. Since the Figure is in color already the authors might consider changing to make it easier for the reader to distinguish different conditions.

We have reformatted the figures with different colors as suggested by the Reviewer.

References

1. Dardenne E, Beltran H, Benelli M, Gayvert K, Berger A, Puca L, et al. N-Myc Induces an EZH2-Mediated Transcriptional Program Driving Neuroendocrine Prostate Cancer. *Cancer cell* 2016;30(4):563-77 doi 10.1016/j.ccell.2016.09.005.
2. Chen L, Alexe G, Dharia NV, Ross L, Iniguez AB, Conway AS, et al. CRISPR-Cas9 screen reveals a MYCN-amplified neuroblastoma dependency on EZH2. *J Clin Invest* 2018;128(1):446-62 doi 10.1172/JCI90793.
3. Baluapuri A, Wolf E, Eilers M. Target gene-independent functions of MYC oncoproteins. *Nature reviews* 2020;21(5):255-67 doi 10.1038/s41580-020-0215-2.
4. Otto T, Horn S, Brockmann M, Eilers U, Schuttrumpf L, Popov N, et al. Stabilization of N-Myc is a critical function of Aurora A in human neuroblastoma. *Cancer cell* 2009;15(1):67-78 doi 10.1016/j.ccr.2008.12.005.
5. Richards MW, Burgess SG, Poon E, Carstensen A, Eilers M, Chesler L, et al. Structural basis of N-Myc binding by Aurora-A and its destabilization by kinase inhibitors. *Proceedings of the National Academy of Sciences of the United States of America* 2016;113(48):13726-31 doi 10.1073/pnas.1610626113.

We have cited all these references in the revised manuscript.

Reviewer #2, with expertise in EZH2 and MYC and mouse models:

This is a very interesting study focused on analyzing regulators of MYC transcription factor functions through protein-protein interactions. The authors unveil a catalytic-independent function of EZH2 in the stability of MYC-transcription factors. The manuscript is well written and a logical set of experiments is conducted and properly controlled. Multiple methodological approaches (shRNA, CRISPR, pharmacological inhibition) support the robustness of their findings.

We appreciate the Reviewer's expert comments.

However, some questions remain open and need clarification before the manuscript gets accepted.

Major concerns:

1. The authors claim in the title that EZH2 controls MYC stability in general. This conclusion is withdrawn from experiments carried out in different cancer models where only c-Myc is expressed (Daudi and NCI-H2171 cells) or in models where N-Myc is expressed (Kelly cells). However, most of Neuroblastoma cells co-express N-MYC and C-MYC. Thus, it will be important to distinguish whether EZH2 depletion affects both c-MYC and/or N-Myc in neuroblastoma cells. Thus, the question is, in cells where N-Myc and C-Myc are expressed, EZH2 depletion has the same functional consequences? And more importantly, the EZH2 depletion effects could be rescued by only overexpressing c-MYC or N-MYC?

The Reviewer raised an excellent scientific question. We performed EZH2 depletion in multiple *MYCN*-amplified and -nonamplified neuroblastoma cell lines. Intriguingly, depletion of EZH2 expression by specific sgRNAs led to a marked decline in *MYCN* protein abundance in both *MYCN*-amplified Kelly, SK-N-BE2, and SMS-KAN cells and *MYCN*-nonamplified NBLS cells which express high levels of *MYCN* (revised Fig. 2a). Since most *MYCN*-nonamplified neuroblastoma cells exhibited high MYC levels, we also inhibited EZH2 expression in SK-N-AS, SH-SY5Y and SK-N-SH cells. As expected, EZH2 inhibition caused a significant depletion of MYC expression in all these cell lines (revised Fig. 2a and Supplementary Fig. 2a). Of note, SK-N-SH cells express both high levels of MYC and low levels of *MYCN*, EZH2 depletion decreased both MYC and *MYCN* expression (revised Supplementary Fig. 2a).

These findings prompted us to examine whether similar regulation occurs in other MYC(N) driven tumor cells. To evaluate this, we performed EZH2 depletion in representative MYC(N)-driven small cell carcinoma NCI-H526 and NCI-H82 cells as well as MYC-driven Burkitt's lymphoma Daudi cells. As expected, depletion of EZH2 expression by specific sgRNAs significantly reduced MYC(N) protein abundance in all these cells (revised Fig. 7a-c). In addition, we depleted EZH2 in non-small cell lung carcinoma A549 cells and colorectal carcinoma HCT116 cells, both of which exhibit deregulated MYC expression, and EZH2 depletion also caused a significant decrease in MYC abundance (data not shown). All these data suggest that stabilization MYC(N) by EZH2 is a general phenomenon.

Please kindly note that ectopic expression of *MYCN* T50A mutant, which abolished GSK3 β -mediated phosphorylation and circumvented *MYCN* degradation by proteasome, significantly reversed growth inhibition of *MYCN*-amplified Kelly cells by either knockdown of endogenous EZH2 or administration of DZNep (revised Fig. 5c, d). Similar results were obtained in MYC-driven SK-N-AS cells, where ectopic expression of the MYC T58A mutant also significantly reversed growth inhibition upon EZH2 knockdown (revised Supplementary Fig. 5d). In contrast, ectopic expression of SUZ12 (another PRC2 core subunit) failed to rescue *MYCN* expression and cell proliferation resulting from knockdown of endogenous EZH2 (revised Supplementary Fig. 5a), suggesting a specific role of MYC(N) in this event.

2. MYCN-amplification is a genetic abnormality that dictates poor prognosis in Neuroblastoma and also other embryonal tumors. However, some of these tumors are also capable of sustaining high MYCN levels without a genetic alteration. The effect of EZH2 on the stability of MYCN are also happening in non-MYCN amplified cell lines?

The Reviewer raised an excellent scientific question. Indeed, depletion of EZH2 expression by specific sgRNAs led to a marked decline in MYCN protein abundance in MYCN-nonamplified NBL cells which express high levels of MYCN (revised Fig. 2a). Addition of MG132 almost completely rescued the MYCN loss caused by EZH2 depletion (revised Supplementary Fig. 3b), arguing that EZH2 promotes MYCN stabilization. Time-course analysis revealed that sgRNA-mediated depletion of EZH2 significantly shortened the half-life of endogenous MYCN in NBL cells (revised Supplementary Fig. 3c), supporting that EZH2 sustains MYCN protein stability in MYCN-nonamplified cells.

3. Conversely, there are some neuroblastoma cell lines that do not express MYCN (e.g. SH-SY5Y). To what extent this lack of MYCN expression can be EZH2-dependent? Please discuss and test whether EZH2 is sufficient to stabilize MYCN in cell lines with low or absent levels of MYCN.

Please kindly note that most MYCN-nonamplified neuroblastoma cells exhibited high MYC levels, including SH-SY5Y cells. Indeed, EZH2 inhibition caused a significant depletion of MYC expression in these cells (revised Fig. 2a), but not in SHEP cells lacking either MYC or MYCN expression (Supplementary Fig. 2b). These results also suggest that EZH2 can hardly impact MYC(N) stabilization if it is not expressed.

4. It seems clear that EZH2 depletion results in a reduction on MYCN levels. When EZH2 is depleted, what are the consequences to other key elements of the PRC2 complex? Can the levels of MYCN be restored (after EZH2 depletion) by the ectopic expression of other PRC2 components (e.g. SUZ12)? Please discuss.

Consistent with previous data (Tan et al, Genes Dev. 2007), EZH2 depletion also reduced the PRC2 subunit SUZ12 and EED levels (Supplementary Fig. 2c). Most likely, depletion of EZH2 caused SUZ12 and EED degradation due to PRC2 destruction.

Please kindly note that ectopic expression of SUZ12 (another PRC2 core subunit) failed to rescue MYCN expression and cell proliferation resulting from knockdown of endogenous EZH2 (revised Supplementary Fig. 5a), suggesting a specific role of EZH2 in this event.

5.-Regarding the correlation between EZH2 and MYCN protein levels in neuroblastoma, is this correlation maintained in all analyzed stages, or in patients with and without MYCN-amplification? Please, clarify. Is any of these correlations also observed at the mRNA level? Please, discuss.

The Reviewer raised a critical scientific question. Since we do not have all the detailed information (*MYCN*-amplification status, disease stage, etc) of the human samples tested, we removed these patient sample data from the revised manuscript based on the expert opinion from Reviewer #3 (minor comment #5 from Reviewer #3).

6.-Regarding the in vivo experiments with anti-PD1, the authors should demonstrate whether EZH2 reduction in vivo, causes an increase in the expression of PD1 (by immunohistochemistry) and how this impact in the infiltration of immune cells in the tumor. This experiments will help to define and characterize the effectors of this observe immune response.

The Reviewer raised a critical scientific question. According to the expert opinion from the other reviewers, we have removed this part from the revised manuscript. We are currently investigating these important issues.

7.-Since most of the functional analyses have been performed on EZH2 and MYCN in Neuroblastoma models, this should be reflected in the title. Although, the results points towards a general mechanism on MYC factors, the authors do not prove it at all levels.

We have edited the title as suggested. As discussed above, we have tested a number of MYC(N) driven tumor cell lines from different cancer types. All these data suggest that stabilization of MYC(N) by EZH2 is a general phenomenon.

Minor concerns:

-Molecular weight bands should be indicated in the immunoblots.

Molecular weight bands have been indicated in the immunoblots as suggested.

Reviewer #3, with expertise in neuroblastoma immunotherapies:

This is a well-written paper with a plethora of data that adds to the information on the interaction of EZH2 with MYC. The novel message in this paper is that EZH2 interaction with MYC is through non-canonical methods that are independent of its methyltransferase activity. The observation that EZH2 has functions other than those related to methyltransferase activity has been previously reported, though its specific interactions with MYC have been incompletely evaluated thus far.

We appreciate the Reviewer's expert comments.

Major Concerns:

1. The majority of experiments are related to NMYCN and not CMYC. The large amount of presented data makes it confusing at times to follow which experiments are related to NMYC and CMYC. Strongly recommend that the paper focus on NMYC. Almost identical experiments (though fewer) were performed to assess EZH2-CYMC interactions. Suggest these be summarized in 1-2 sentences after each result section and all data presented as

Supplementary Files.

We are sorry for this inconvenience. We have extensively revised the whole manuscript based on the Reviewer's suggestion.

2. In this same vein, throughout the paper, the authors should specify if the presented results refer to C-MYC or N-MYC. There are numerous areas in the paper where this is not clearly specified.

We have edited the nomenclature trying to make it easier to follow. Now in the revised manuscript, MYC and MYCN correspond to the protein products of *MYC* and *MYCN* oncogenes, respectively. We are sorry for this inconvenience.

3. A major point that has NOT been addressed by the investigators is the difference between MYCN overexpression and amplification in neuroblastoma. Most of the initial elegant experiments have been carried out in a single MYCN-amplified cell line (Kelly). This has significant implications in the translational analysis of the results. Do EZH2-N-MYC interactions depend on amplification, overexpression or "normal" expression of MYCN. At least some of the initial experiments should be repeated in MYCN-non-amplified cell line to address this issue.

The Reviewer raised a critical scientific question. We performed EZH2 depletion in a number of *MYCN*-amplified and -nonamplified neuroblastoma cell lines. Intriguingly, depletion of EZH2 expression by specific sgRNAs led to a marked decline in MYCN protein abundance in both *MYCN*-amplified Kelly, SK-N-BE2, and SMS-KAN cells and *MYCN*-nonamplified NBL cells which express high levels of MYCN (revised Fig. 2a). Since most *MYCN*-nonamplified neuroblastoma cells exhibited high MYC levels, we also inhibited EZH2 expression in SK-N-AS, SH-SY5Y and SK-N-SH cells. As expected, EZH2 inhibition caused a significant depletion of MYC expression in all these cell lines (revised Fig. 2a and Supplementary Fig. 2a). Of note, SK-N-SH cells express both high levels of MYC and low levels of MYCN, EZH2 depletion decreased both MYC and MYCN expression (revised Supplementary Fig. 2a).

These findings prompted us to examine whether similar regulation occurs in other MYC(N) driven tumor cells. To evaluate this, we performed EZH2 depletion in representative MYC(N)-driven small cell carcinoma NCI-H526 and NCI-H82 cells as well as MYC-driven Burkitt's lymphoma Daudi cells. As expected, depletion of EZH2 expression by specific sgRNAs significantly reduced MYC(N) protein abundance in all these cells (revised Fig. 7a-c). In addition, we depleted EZH2 in non-small cell lung carcinoma A549 cells and colorectal carcinoma HCT116 cells, both of which exhibit deregulated MYC expression, and EZH2 depletion also caused a significant decrease in MYC abundance (data not shown). All these data suggest that stabilization of MYC(N) by EZH2 is a general phenomenon.

4. The choice of human cell lines for the various experiments also seems somewhat arbitrary. The characteristics of various cell lines should be described and the rationale for their choice explained. (It is only on line 276 that SHEP MYCN-ER is characterized in passing).

As discussed above, we have tested a number of MYC(N) driven tumor cell lines from different cancer types. All these data suggest that stabilization MYC(N) by EZH2 is a general phenomenon. We have also described the characteristics of these representative tumor cell lines, which are commonly used by colleagues world-wide, in the revised manuscript.

5. Data from human subjects has been included, but no mention has been made about human subject protection e.g. IRB approval.

The Reviewer raised a critical scientific question. Since we do not have all the detailed information (*MYCN*-amplification status, disease stage, etc) of the human samples tested, we removed these patient sample data from the revised manuscript based on the Reviewer's expert comment.

6. The rationale for considering EZH2 inhibition in combination with immune checkpoint blockade (Lines 302-312) has been very cursorily developed. A major reason for the lack of success for this immunotherapeutic approach in NB has been the lack of HLA Class expression in addition to the PD1/PDL1 axis. Similarly lines 296-301 refer to PD-L1 mRNA not protein. In general, the interpretation of data to emphasize a role for EZH2 inhibition for enhancing immunotherapy is not convincing. Recommend that this section be removed from this paper especially since the more established role of EZH2 function in immune effector cells that would need utilization in such a strategy has not been explored.

The Reviewer raised a critical scientific question. According to the expert opinion from the Reviewer, we have removed this part from the revised manuscript. We are currently in vestigating these important issues.

7. Would be more valuable in the Discussion section to focus on the mechanisms of EZH2-MYC interactions uncovered by the investigators rather than to discuss the potential for immunotherapeutic strategies.

We have extensively edited the whole manuscript based on the Reviewer's comment.

Minor

1. Last paragraph of introduction is more suited to discussion section.

We have edited this paragraph based on the Reviewer's comment.

2. Consider expanding figure legends to concisely state the results of the experiment rather than methods alone.

We have edited the figure legends based on the Reviewer's comment.

3. Line 197: clarify what ectopic expression means in this context.

The sentence of "Ectopic expression of EZH2 WT or H689A alone..." means ectopic expression of EZH2 WT or H689A mutant in control shRNA treated Kelly cells (line 265 in the revised text). The purpose of these experiments is to examine whether ectopic expression of either EZH2 WT or H689A mutant effected proliferation of the control shRNA treated Kelly cells.

4. Line 228: the statement about off-target effects of GSK 126 is speculative and should either be deleted or moved to Discussion

We have deleted this statement as suggested by the Reviewer.

5. Line 265-268; 298-299: These data are cursory and no further detail is available on the human samples tested. Recommend removing data on human tissue samples from this paper.

The Reviewer raised a critical scientific question. We have removed this part from the revised manuscript based on the Reviewer's comment.

Reviewer #4, with expertise in PRC2/EZH2:

The study by e uncovered a new mechanism of EZH2 in promoting tumorigenesis in which EZH2 acts through interaction with Myc and maintains its stability. The study is very interesting as it shows that this activity of EZH2 does not require its methyltransferase activity. The mechanism is well elucidated with a rigor set of experiments showing EZH2 induces MYC stability through antagonizing MYC ubiquitination. As such small molecules of EZH2 inhibitor such as DZNep, which causes EZH2 degradation as a whole, can deplete MYC protein expression and inhibits MYC-driven cancer cell proliferation. In contrast, the catalytic inhibitors of EZH2 are unable to do this. This finding has important therapeutic implications, as currently, there are no targeted therapies against Myc. The study further provides novel mechanistic insights into the role of EZH2 in tumorigenesis and highlights the importance of developing EZH2 degraders for cancer therapy. The manuscript is well written, and the experiments are well executed with proper controls. Impressively, both in vitro and in vivo models and data from clinical samples have been provided to support the claim.

We appreciate the Reviewer's expert comments.

Major Comments

The last part shows a role of EZH2/MYC in suppressing PDL1. As such, depletion of EZH2 by DZNep is shown to restore PDL1 expression and thus sensitize anti PDL1 treatment. Although this finding is interesting and potentially expands the role of EZH2/MYC in immune regulation, the mechanism for PDL1 expression might be more complex that may involve both canonical

histone and non-histone activities. Several questions remain, eg does EZH2 suppress PDL1 expression through histone methylation at PDL1 loci ? Have the enzymatic inhibitors of EZH2 GSK126 been tested in this context? Does H3K27me3 at PDL1 locus show any changes before and after DZNep treatment?

The Reviewer raised a critical scientific question. According to the expert opinion from the Reviewer, we have removed this part from the revised manuscript. We are currently investigating these important issues.

I feel that the data obtained from Figures 1-5 have adequately delivered sufficient novelty and robustness for publication. This review suggests the authors consider taking out the last part of PDL1, which can be further investigated in a separate study.

We appreciate the Reviewer's expert comments. We have removed this part from the revised manuscript. Meanwhile, we have extensively revised the whole manuscript based on the comments from all 4 Reviewers.

Minor comments

Some key references about DZNep and methyltransferase-independent role of EZH2 should be included:

Tan et al., *Genes Develop* 2007, which reported DZNep as the first-in-class EZH2 inhibitor that induce EZH2 degradation. /doi/10.1101/gad.1524107.

Lee et al, *Molecule Cell*, 2011, which is among the first to report non-catalytic activity of EZH2 in cancer <https://doi.org/10.1016/j.molcel.2011.08.011>

We have included these important references as suggested by the Reviewer. We are sorry for this inadvertent mistake.

In summary, we hope that these changes have adequately addressed all the concerns and that the manuscript is now acceptable for publication in *Nature Communications*. Thank you very much for your time, effort, and great help in this matter.

Reviewers' Comments:

Reviewer #1:

Remarks to the Author:

In my view, the authors have come back with a much improved version of the manuscript and have responded well to all major criticisms that I raised in my first review. A few issues remain: for example, the authors show that MycBoxII binds EzH2, but Fbxw7 binds specifically MycBoxI, so how they compete is a bit unclear. Nevertheless, the aggregate of data is now very strong and I support publication of the manuscript.

Reviewer #2:

Remarks to the Author:

The authors have addressed all my concerns.

Reviewer #3:

Remarks to the Author:

In this revised manuscript the authors have made substantial changes. These include:

1. Expansion of number of cell lines tested
2. Extension of their findings to a second cancer type
3. Performed additional experiments to investigate the direct interaction of EZH2 with MYC
4. Clarified and reformatted the presentation of some data.
5. Removed some unsubstantiated statements including references to role for immunotherapy.

Reviewer #4:

Remarks to the Author:

The authors have addressed my concern and I have no further comments

The following is a point-by-point response to the comments.

Reviewer #1 (Remarks to the Author):

In my view, the authors have come back with a much improved version of the manuscript and have responded well to all major criticisms that I raised in my first review. A few issues remain: for example, the authors show that MycBoxli binds EzH2, but Fbxw7 binds specifically MycBoxl, so how they compete is a bit unclear. Nevertheless, the aggregate of data is now very strong and I support publication of the manuscript.

We appreciate the Reviewer's expert comments.

Reviewer #2 (Remarks to the Author):

The authors have addressed all my concerns.

We appreciate the Reviewer's expert comments.

Reviewer #3 (Remarks to the Author):

In this revised manuscript the authors have made substantial changes. These include:

1. Expansion of number of cell lines tested
2. Extension of their findings to a second cancer type
3. Performed additional experiments to investigate the direct interaction of EZH2 with MYC
4. Clarified and reformatted the presentation of some data.
5. Removed some unsubstantiated statements including references to role for immunotherapy.

We appreciate the Reviewer's expert comments.

Reviewer #4 (Remarks to the Author):

The authors have addressed my concern and I have no further comments.

We appreciate the Reviewer's expert comments.

In summary, we hope that the manuscript is now acceptable for publication in *Nature Communications*. Thank you very much for your time, effort, and great help in this matter.